# Heterogeneous efflux pump expression underpins phenotypic resistance to antimicrobial peptides

Ka Kiu Lee[1,2], Urszula Łapińska[1,2], Giulia Tolle[3], Maureen Micaletto[1,2], Bing Zhang[4], Wanida Phetsang[4], Anthony D Verderosa[4], Brandon M Invergo[5], Joseph Westley[6], Attila Bebes[2,7], Raif Yuecel[2,7], Paul A O'Neill[2,8], Audrey Farbos[2,8], Aaron R Jeffries[2,8], Stineke van Houte[6], Pierluigi Caboni[3], Mark AT Blaskovich[4,9], Benjamin E Housden[1,10], Krasimira Tsaneva-Atanasova[1,11], Stefano Pagliara[1,2]*

[1]Living Systems Institute, University of Exeter, Exeter, United Kingdom; [2]Biosciences, University of Exeter, Exeter, United Kingdom; [3]Department of Life and Environmental Sciences, University of Cagliari, Cittadella Universitaria, Cagliari, Italy; [4]Centre for Superbug Solutions, Institute for Molecular Bioscience, The University of Queensland, St. Lucia, Brisbane, Australia; [5]Process Integration and Predictive Analytics, PIPA LLC, Davis, United States; [6]Centre for Ecology and Conservation and Environment and Sustainability Institute, University of Exeter, Exeter, United Kingdom; [7]Exeter Centre for Cytomics, Henry Wellcome Building for Biocatalysis, Biosciences, University of Exeter, Exeter, United Kingdom; [8]Exeter Sequencing Service, Biosciences, University of Exeter, Exeter, United Kingdom; [9]ARC Training Centre for Environmental and Agricultural Solutions to Antimicrobial Resistance (CEAStAR), Institute for Molecular Bioscience, The University of Queensland, St. Lucia, Brisbane, Australia; [10]Department of Clinical and Biomedical Sciences, University of Exeter, Exeter, United Kingdom; [11]Department of Mathematics and Statistics, University of Exeter, Exeter, United Kingdom

*For correspondence:
s.pagliara@exeter.ac.uk

## eLife Assessment

This **important** study by Lee et al. explores the heterogeneous response of non-growing bacteria to the antimicrobial peptide (AMP) tachyplesin. The authors identify a subpopulation of cells that evade lethal damage by limiting the intracellular accumulation of a fluorescently labeled tachyplesin analog. The study provides **compelling** evidence that reduced drug accumulation underlies the decreased susceptibility of this subpopulation to the AMP. The molecular basis of this phenotype is well supported by the data.

**Abstract** Antimicrobial resistance threatens the viability of modern medical interventions. There is a dire need to develop novel approaches to counter resistance mechanisms employed by starved or slow-growing pathogens that are refractory to conventional antimicrobial therapies. Antimicrobial peptides have been advocated as potential therapeutic solutions due to the low levels of genetic resistance observed in bacteria against these compounds. However, here we show that subpopulations of stationary phase *Escherichia coli* and *Pseudomonas aeruginosa* survive tachyplesin treatment without acquiring genetic mutations. These phenotypic variants display enhanced efflux activity to limit intracellular peptide accumulation. Differential regulation of genes involved in outer membrane vesicle secretion, membrane modification, and protease activity was also observed

between phenotypically resistant and susceptible cells. We discovered that the formation of these phenotypic variants could be prevented by administering tachyplesin in combination with sertraline, a clinically used antidepressant, suggesting a novel approach for combatting antimicrobial-refractory stationary phase bacteria.

## Introduction

Antimicrobial resistance (AMR) was estimated to claim 1.27 million lives in 2019 (*Murray et al., 2022*) and is predicted to claim 10 million lives annually by 2050 (*Naghavi et al., 2024*). Besides the deaths caused directly by infections, AMR also threatens the viability of many modern medical interventions, such as surgery and organ transplant. This challenging situation has motivated the development of strategies to combat resistance mechanisms utilised by bacterial pathogens, particularly when they enter states of dormancy or slow growth and become more refractory to conventional antimicrobial treatments (*Tamer and Toprak, 2017*; *Ahmad et al., 2025*). In recent decades, natural defences such as bacteriocin proteins, peptides, endolysins, and antibodies have garnered substantial attention as potential clinical antimicrobial agents (*MacNair et al., 2024*). Antimicrobial peptides (AMPs) have been advocated as potential therapeutic solutions to the AMR crisis. Notably, polymyxin-B and -E are in clinical use, and additional AMPs are undergoing clinical trials (*Rima et al., 2021*). AMPs can simultaneously target different cellular subsystems: they target the double membrane of Gram-negative bacteria by interacting with negatively charged membrane lipids while simultaneously disrupting cytoplasmic processes such as protein synthesis, DNA replication, cell wall biosynthesis, metabolism, and cell division (*Blair et al., 2022*).

Bacteria have developed genetic resistance to AMPs, including proteolysis by proteases, modifications in membrane charge and fluidity to reduce affinity, and extrusion by AMP transporters. However, compared to small molecule antimicrobials, AMP resistance genes typically confer smaller increases in resistance in experimental evolution analyses, with polymyxin-B and CAP18 being notable exceptions (*Spohn et al., 2019*). Moreover, mobile resistance genes against AMPs are relatively rare and horizontal acquisition of AMP resistance is hindered by phylogenetic barriers owing to functional incompatibility with the new host bacteria (*Kintses et al., 2019b*). Plasmid-transmitted polymyxin resistance constitutes a notable exception (*Nang et al., 2019*), possibly because polymyxins are the only AMPs that have been in clinical use to date (*Kintses et al., 2019b*). However, whether bacterial populations harbour subpopulations that are transiently phenotypically resistant to AMPs without the acquisition of genetic mutations remains an open question. While there is limited understanding regarding AMPs and transient phenotypic resistance (*Kepiro et al., 2020*), this phenomenon has been widely investigated in the case of small molecule antimicrobials and bacteriophages (*Bamford et al., 2017*; *Attrill et al., 2021*; *Attrill et al., 2023*; *Bollen et al., 2023*; *Urbaniec et al., 2022*; *Campey et al., 2024*). Phenotypic resistance to antimicrobials is known to result in negative clinical outcomes and contributes to the emergence of genetic resistance (*Van den Bergh et al., 2017*; *Brandis et al., 2023*; *Barrett et al., 2019*; *Windels et al., 2019a*). Therefore, further investigation into understanding the initial trajectories of the emergence of AMP resistance is required for the development of affordable, safe, and effective AMP treatments while simultaneously circumventing the pitfalls that have fuelled the current AMR crisis (*Jangir et al., 2021*; *Lazzaro et al., 2020*).

The 17 amino acid cationic β-hairpin AMP tachyplesin (2.27 kDa) (*Nakamura et al., 1988*) is a promising candidate AMP because it displays broad-spectrum, potent antibacterial and antifungal efficacy, minimal haemolytic effect, and minimal evolution of bacterial resistance (*Spohn et al., 2019*). It has been speculated that tachyplesin simultaneously targets multiple cellular components (*Liu et al., 2018*; *Imura et al., 2007*). However, further investigation is needed to fully understand this aspect, along with exploring bacteria's potential to transiently survive tachyplesin exposure without acquiring genetic mutations. Understanding the emergence of phenotypic resistance to tachyplesin could provide valuable insights into optimising the therapeutic potential of tachyplesin and other AMPs against microbial threats.

Achieving inhibitory concentrations proximal to its cellular target is crucial for tachyplesin (and antimicrobials in general) efficacy (*Rybenkov et al., 2021*). At low concentrations, AMPs induce the formation of transient pores in bacterial membranes that allow transmembrane conduction of ions but not leakage of intracellular molecules (*Rima et al., 2021*; *Blair et al., 2022*). Beyond a critical ratio

between the AMP and membrane lipid concentrations, these pores become stable, leading to leakage of cellular content, loss of transmembrane potential, and eventual cell death (*Rima et al., 2021*; *Blair et al., 2022*). Moreover, it is conceivable that tachyplesin, and other AMPs, must penetrate the bacterial inner membrane and accumulate at intracellular concentrations sufficient to interfere with cellular targets (*Silver, 2016*). Simultaneously, tachyplesin must avoid efflux via protein pumps, such as the *Escherichia coli* and *Yersinia pestis* tripartite pumps AcrAB-TolC and EmrAB-TolC, that confer genetic resistance against AMPs like protamine and polymyxin-B (*Weatherspoon-Griffin et al., 2014*; *Lister et al., 2012*).

Crucially, recent discoveries have unveiled an association between reduced antimicrobial accumulation and transient phenotypic resistance to antimicrobials (*Pu et al., 2016*; *Stone et al., 2020*; *Whittle et al., 2021*; *El Meouche and Dunlop, 2018*; *Łapińska et al., 2022*). Therefore, we hypothesise that phenotypic variations in the mechanisms regulating membrane lipid and protein compositions allow bacterial subpopulations to shift the balance between influx and efflux, reduce the intracellular accumulation of tachyplesin, and survive treatment without the emergence of genetic mutations.

To test this hypothesis, we investigated the accumulation and efficacy of tachyplesin-1 against individual *E. coli*, *Pseudomonas aeruginosa*, *Klebsiella pneumoniae,* and *Staphylococcus aureus* cells in their stationary phase of growth. To determine the mechanisms underpinning phenotypic resistance to tachyplesin, we performed comparative gene expression profiling and membrane lipid abundance analysis between subpopulations susceptible and transiently resistant to tachyplesin treatment. We used this knowledge to design a novel drug combination that increases tachyplesin efficacy in killing non-growing bacteria that are notoriously more refractory to treatment with AMPs or other antimicrobials. Our data demonstrate that streamlining antibacterial drug discovery could be facilitated by addressing the fundamental issue of low intracellular drug accumulation. We propose new combination treatments as a promising foundation for developing peptide-based drug combinations against dormant or slow-growing bacteria.

## Results

### Tachyplesin accumulates heterogeneously within clonal *E. coli* and *P. aeruginosa* populations

To investigate the accumulation of tachyplesin in individual bacteria, we utilised a fluorescent derivative of the antimicrobial peptide, tachyplesin-1-nitrobenzoxadiazole (tachyplesin-NBD) (*Blaskovich et al., 2019*) in combination with a flow cytometry assay adapted from previously reported protocols (*Zhang et al., 2023b*; *Zhou et al., 2015*; *Figure 1—figure supplement 1* and Methods).

As expected, we found that bacterial fluorescence, due to accumulation of tachyplesin-NBD in individual stationary phase *E. coli* BW25113 cells, increased an order of magnitude when the extracellular concentration of tachyplesin-NBD increased from 0 to 8 µg mL⁻¹ (0–3.2 µM). At the latter concentration, the single-cell fluorescence distribution displayed a median value of 2400 a.u. after 60 min treatment at 37 °C (*Figure 1A*). However, a further increase in the extracellular concentration of tachyplesin-NBD to 16 µg mL⁻¹ (6.3 µM) revealed a noticeable bimodal distribution in single-cell fluorescence. We classified bacteria in the distribution with a median value of 2400 a.u. as low accumulators and those in the distribution with a median value of 28,000 a.u. as high accumulators (blue and red shaded areas, respectively, in *Figure 1A* and S1D-J).

A further increase in the extracellular concentration of tachyplesin-NBD to 46 µg mL⁻¹ (18.2 µM) still yielded a bimodal distribution in single-cell fluorescence, 43% of the population being low accumulators; in contrast, only 5% of the population were low accumulators when tachyplesin-NBD treatment at 46 µg mL⁻¹ (18.2 µM) was carried out against exponential phase *E. coli* (*Figure 1—figure supplement 2*). Notably, the minimum inhibitory concentration (MIC) of tachyplesin-NBD against exponential phase *E. coli* is 1 µg mL⁻¹ (0.4 µM) (*Łapińska et al., 2022*). Therefore, our data suggest that even at extracellular concentrations surpassing growth-inhibitory levels, stationary phase *E. coli* populations harbour two distinct phenotypes with differing physiological states that respond to treatment differently with high accumulators displaying over 10-fold greater fluorescence than low accumulators.

Subsequently, we tested whether this newly observed bimodal distribution of tachyplesin-NBD accumulation was unique to the *E. coli* BW25113 strain. We found that the bimodal distribution in single-cell fluorescence was also present in nine *E. coli* clinical isolates (treated with tachyplesin-NBD

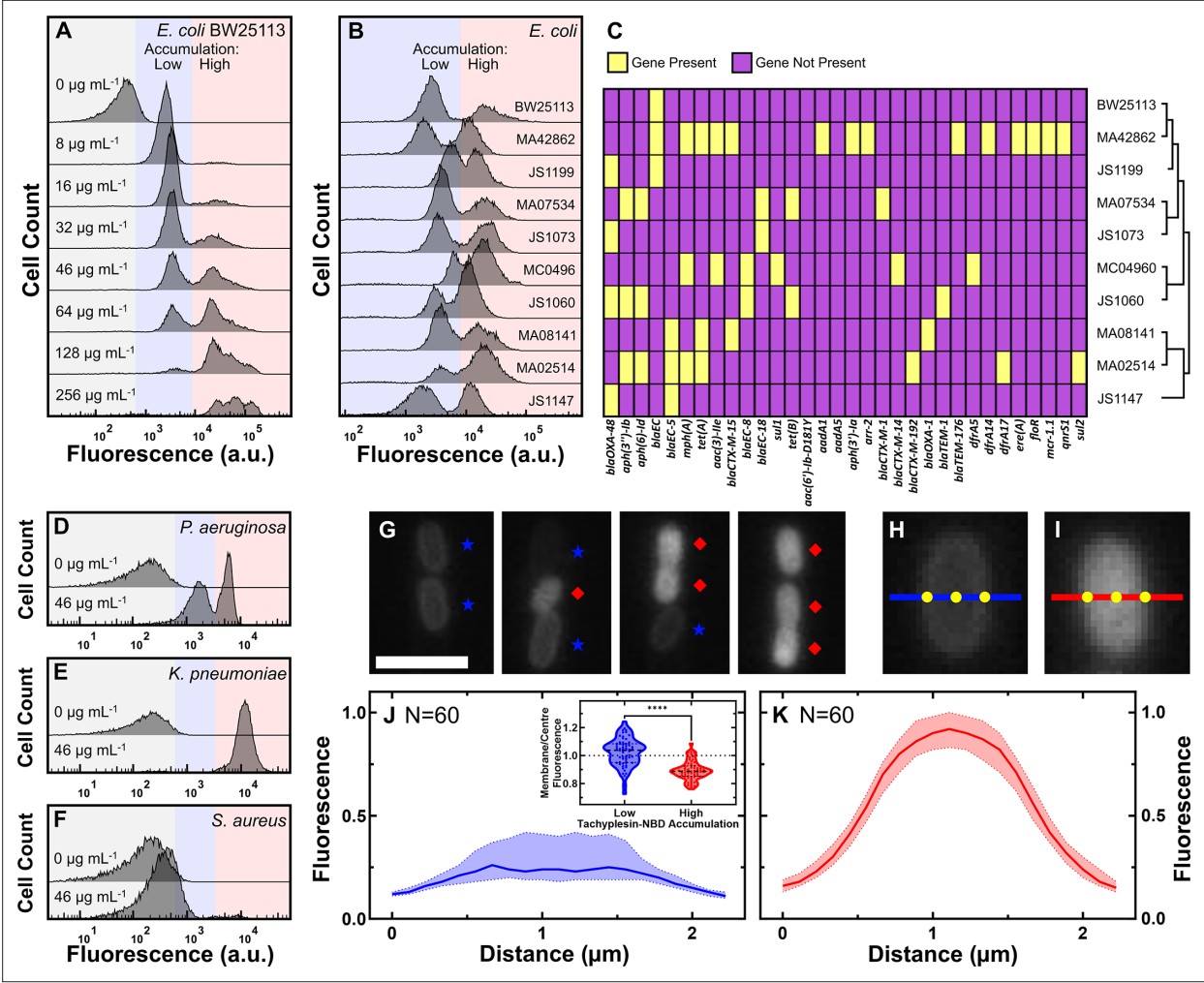

**Figure 1.** Tachyplesin accumulates heterogeneously within clonal *E. coli* and *P. aeruginosa* populations. (**A**) Tachyplesin-nitrobenzoxadiazole (NBD) accumulation in stationary phase *E. coli* BW25113 treated with increasing concentrations of tachyplesin-NBD for 60 min (0, 8, 16, 32, 46, 64, 128, and 256 µg mL⁻¹ or 0.0, 3.2, 6.3, 12.7, 18.2, 25.4, 50.7, and 101.5 µM). Shaded regions show low (blue) and high (red) tachyplesin-NBD accumulation within an isogenic *E. coli* BW25113 population. Each histogram reports 10,000 recorded events and is a representative example of accumulation data from three independent biological replicates (*Figure 1—figure supplement 1*). (**B**) Tachyplesin-NBD accumulation in nine stationary phase *E. coli* clinical isolates treated with 46 µg mL⁻¹ (18.2 µM) tachyplesin-NBD for 60 min. Accumulation data for *E. coli* BW25113 is reproduced from panel A. (**C**) Heatmap showing the presence of antimicrobial resistance (AMR) genes and phylogenetic data of the clinical isolates. Corresponding gene products and the antimicrobials these genes confer resistance to are reported in *Figure 1—source data 2*. (**D–F**) Tachyplesin-NBD accumulation in *P. aeruginosa* (**D**), *K. pneumoniae* (**E**), and *S. aureus* (**F**) treated with 0 or 46 µg mL⁻¹ (0 or 18.2 µM) tachyplesin-NBD for 60 min. Each histogram in each panel shows a representative example of accumulation data from three independent biological replicates. (**G**) Representative fluorescence images depicting low and high tachyplesin-NBD accumulators within an *E. coli* BW25113 population. Bacteria were continuously exposed to 46 µg mL⁻¹ (18.2 µM) tachyplesin-NBD for 60 min in a microfluidic mother machine device. Blue stars and red diamonds indicate low and high accumulators, respectively. Scale bar: 3 µm. (**H, I**) Representative fluorescence images of individual low (**H**) and high (**I**) tachyplesin-NBD accumulators, respectively. Blue and red lines show a 2.2 µm-long cross-sectional line used for measuring fluorescence profile values in J and K, with the origin on the left side. (**J, K**) Normalised median (solid line), lower and upper quartiles (dotted lines) of fluorescence profile values of *E. coli* BW25113 cells plotted against the distance along the left side origin of a 2.2-µm-long straight line in low (**J**) and high (**K**) tachyplesin-NBD accumulators, respectively. Phenotype assignment was further verified via propidium iodide staining (see *Figure 2G*). Inset: each dot represents the corresponding membrane-to-cell centre fluorescence ratio measured at the points indicated in H and I, dashed lines indicate the median and quartiles of each distribution. Statistical significance was assessed using an unpaired nonparametric Mann-Whitney U test with a two-tailed p-value and confidence level at 95%. ****p<0.0001. Data were collected from three independent biological replicates.

The online version of this article includes the following source data and figure supplement(s) for figure 1:

**Source data 1.** Flow cytometry, genomics, and microscopy data were used to generate the graphs presented in *Figure 1*.

**Source data 2.** Antimicrobial resistance genes within clinical isolates.

*Figure 1 continued on next page*

Figure 1 continued

**Figure supplement 1.** Gating strategy and dose-dependent response to tachyplesin-nitrobenzoxadiazole (NBD) treatment.

**Figure supplement 2.** Tachyplesin-nitrobenzoxadiazole (NBD) accumulation in stationary and exponential phase *E. coli* BW25113.

**Figure supplement 3.** Dose-dependent response to arenicin-nitrobenzoxadiazole (NBD), polymyxin-B-NBD, and octapeptin-NBD treatment.

**Figure supplement 4.** Impact of environmental temperature and proteinase K on tachyplesin-nitrobenzoxadiazole (NBD) accumulation.

**Figure supplement 5.** Distribution of single-cell tachyplesin-nitrobenzoxadiazole (NBD) accumulation in the microfluidic mother machine.

**Figure supplement 6.** Representative live-cell confocal microscopy images of *E. coli* treated with tachyplesin-nitrobenzoxadiazole (NBD) and FM 4–64 FX.

**Figure supplement 7.** Comparison of cell sizes between low and high tachyplesin-nitrobenzoxadiazole (NBD) accumulators.

at 46 µg mL$^{-1}$ or 18.2 µM for 60 min) characterised by a diverse genetic background and various AMR genes, such as the polymyxin-E resistance gene *mcr1* (*Majewski et al., 2021 Figure 1B and C*, *Figure 1—source data 2*). The median fluorescence of low accumulators in strains JS1147 and MC04960 was shifted to the left and to the right, respectively, compared to BW25113, whereas the median fluorescence of high accumulators for strain JS1060 was shifted to the left compared to BW25113. Remarkably, all isolates tested harboured low accumulators of tachyplesin-NBD.

Furthermore, to examine whether the bimodal distribution of tachyplesin-NBD accumulation also occurred in other highly virulent bacterial pathogens, we treated three members of the ESKAPE pathogens with 46 µg mL$^{-1}$ (18.2 µM) tachyplesin-NBD for 60 min while in their stationary phase of growth. In *P. aeruginosa* populations, we observed a bimodal distribution of tachyplesin-NBD accumulation with the presence of low accumulators (*Figure 1D*). In contrast, we did not find evidence of low tachyplesin-NBD accumulators in *K. pneumoniae* (*Figure 1E*) and recorded only low levels of tachyplesin-NBD accumulation in *S. aureus* (*Figure 1F*) in accordance with previous reports (*Edwards et al., 2017*). Finally, we tested whether this newly observed bimodal distribution was a feature unique to tachyplesin or is widespread across different AMPs. We found that *E. coli* BW25113 displayed only low accumulators of another β-hairpin AMP, arenicin-NBD, and only high accumulators of cyclic lipopeptide AMPs, polymyxin-B-NBD and octapeptin-NBD (*Figure 1—figure supplement 3*).

These data demonstrate that two key pathogens display bimodal accumulation of tachyplesin, although only limited evolution of genetic resistance against tachyplesin has been previously observed (*Spohn et al., 2019*), but not towards polymyxin-B against which they develop genetic resistance (*Spohn et al., 2019*).

## Tachyplesin accumulates primarily on the membranes of low accumulators

Next, we set out to characterise the differences between low and high tachyplesin accumulators. First, we found that *E. coli* low accumulators treated with 46 µg mL$^{-1}$ (18.2 µM) tachyplesin-NBD for 60 min at 37 °C exhibited a fluorescence distribution akin to the entire population of cells treated with 46 µg mL$^{-1}$ (18.2 µM) tachyplesin-NBD at 0 °C (*Figure 1—figure supplement 4*). At this low temperature, antimicrobials adhere non-specifically to bacterial surfaces as the passive and active transport across bacterial membranes is significantly diminished (*Zhou et al., 2015*). Furthermore, post-binding peptide-lipid interactions are significantly diminished at 0 °C, but not the primary binding step (*Eckert et al., 2006*; *Lehrer et al., 1985*). Moreover, we found that the fluorescence of high accumulators did not decrease over time when tachyplesin-NBD was removed from the extracellular environment and bacteria were treated with 20 µg mL$^{-1}$ (0.7 µM) proteinase K, a widely occurring serine protease that can cleave the peptide bonds of AMPs (*Bajorath et al., 1988*; *Glibowicka et al., 2022*; *Song et al., 2021*; *Figure 1—figure supplement 4*). These data suggest that tachyplesin-NBD primarily accumulates intracellularly in high accumulators.

Next, we measured tachyplesin-NBD accumulation in individual *E. coli* cells using our microfluidics-based microscopy platform (*Łapińska et al., 2019*; *Cama et al., 2020*; *Glover et al., 2022*) with the well-established mother machine device (*Wang et al., 2010*; *Bergmiller et al., 2017*). The mother machine is equipped with thousands of microfluidic channels that trap and host individual *E. coli* cells, enabling controlled medium exchange via pressure-driven microfluidics. Consistent with our flow cytometry data, we observed a bimodal accumulation of tachyplesin-NBD (*Figure 1—figure supplement 5*), with the presence of low and high accumulators (*Figure 1G–I*, blue stars and red

diamonds, respectively). However, the relative abundance of low accumulators was lower in the micro-fluidics experiments possibly due to lower cell densities. Low accumulators exhibited a significantly higher membrane-to-cell centre fluorescence ratio compared to high accumulators (p-value <0.0001, *Figure 1J and K*) and confocal microscopy further confirmed that tachyplesin-NBD is present only on the cell membranes of low accumulators and primarily intracellularly in high accumulators (*Figure 1— figure supplement 6*).

We also compared the forward scatter and violet side scatter values, and cell lengths between low and high tachyplesin-NBD accumulators. We found that these two phenotypes had similar forward and violet side scatter values (*Figure 1—figure supplement 7*), and high accumulators were signifi-cantly smaller than low accumulators in the microfluidics platform (p-value <0.0001, *Figure 1—figure supplement 7*). These data clearly exclude the possibility that high tachyplesin-NBD fluorescence is attributed to larger cell sizes.

## Tachyplesin accumulation on the bacterial membranes is insufficient for bacterial eradication

To test the hypothesis that the accumulation of tachyplesin on the bacterial membranes may be insuf-ficient for bacterial eradication, we conducted measurements of tachyplesin-NBD accumulation and efficacy in bacterial eradication.

Stationary phase *E. coli* was treated with 46 µg mL$^{-1}$ (18.2 µM) tachyplesin-NBD for 60 min followed by fluorescence-activated cell sorting to separate low and high accumulators, alongside untreated control cells (*Figure 2A, B*, *Figure 2—figure supplement 1*, and Methods). Subsequently, we assessed the survival of each sorted sample via colony-forming unit assays and found that the survival fraction of low accumulators was not significantly different from that measured for untreated cells, whereas the survival fraction of high accumulators was significantly lower (p-value <0.0001, *Figure 2C*).

To investigate the hypothesis that tachyplesin causes less membrane damage to low accumula-tors, we employed our microfluidics-based microscopy platform to simultaneously measure tachy-plesin-NBD accumulation and bacterial membrane integrity via propidium iodide (PI) (*Zhang et al., 2023a*). We found a strong positive semi-log correlation between tachyplesin-NBD fluorescence and PI fluorescence ($r^2$=0.70, p-value <0.0001): all bacteria exhibiting PI staining had a tachyplesin-NBD fluorescence greater than 800 a.u. and were high accumulators characterised by tachyplesin-NBD localisation both on the bacterial membranes and intracellularly; by contrast, low accumulators, where tachyplesin-NBD is primarily localised on the bacterial membranes, did not stain with PI (*Figure 2D–G*).

Finally, we set out to determine the contribution of low accumulators to the overall population survival. We exposed stationary phase *E. coli* to increasing concentrations of tachyplesin-NBD for 60 min, while measuring tachyplesin-NBD accumulation in individual bacteria and the bacterial eradication efficacy of tachyplesin-NBD at the population level. We used the accumulation data to measure the proportion of low accumulators relative to the whole population of bacteria at increasing tachyplesin-NBD concentrations and found a strong positive linear correlation with the population survival fraction ($r^2$=0.98, p-value <0.0001, *Figure 2H*). Taken together with PI staining data indicating membrane damage caused by high tachyplesin accumulation, these data demonstrate that low accu-mulators, which primarily retain tachyplesin-NBD on the bacterial membranes, maintain membrane integrity and strongly contribute to the survival of the bacterial population in response to tachyplesin treatment.

## Enhanced efflux, membrane lipid alterations, and increased secretion of outer membrane vesicles associated with low accumulation of tachyplesin

Next, we set out to investigate the mechanisms underpinning low accumulation of tachyplesin. Following our established protocols (*Smith et al., 2018*; *Goode et al., 2021a*), we performed genome-wide comparative transcriptome analysis between low accumulators, a 1:1 (*v/v*) mixture of low and high accumulators, high accumulators, and untreated stationary phase bacteria that were sorted via fluorescence-activated cell sorting. Principal component analysis revealed clustering of the replicate transcriptomes of low tachyplesin-NBD accumulators, the 1:1 mixture of low and high accumulators, high accumulators, and untreated control cells with a distinct separation between these groups on the PC1 plane (*Figure 3—figure supplement 1*).

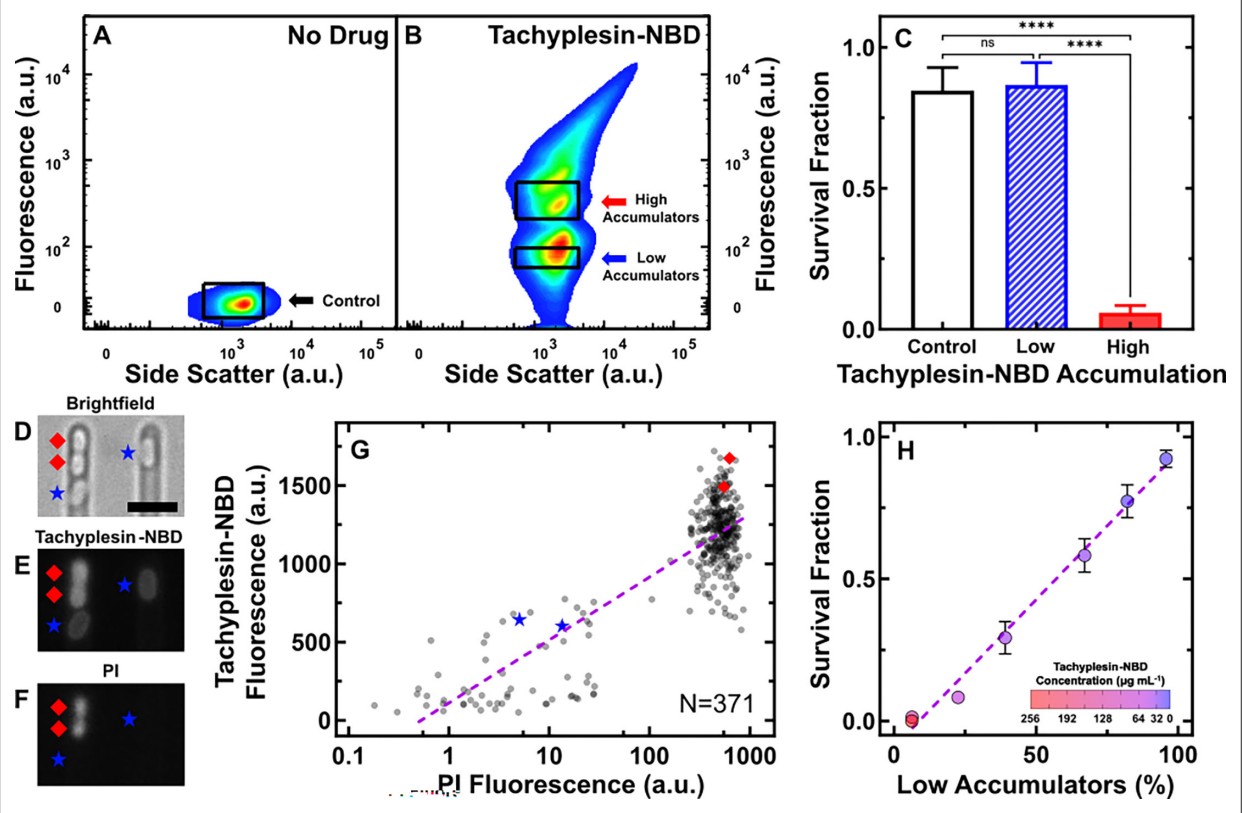

**Figure 2.** Intracellular tachyplesin accumulation is essential for its antimicrobial efficacy. (**A, B**) Fluorescence and side scatter values of individual *E. coli* treated with M9 at 37 °C for 60 min (**A**) and 46 µg mL$^{-1}$ (18.2 µM) tachyplesin-nitrobenzoxadiazole (NBD) in M9 at 37 °C for 60 min (**B**). The black rectangles show the gates used to sort approximately one million untreated control cells, low and high tachyplesin-NBD accumulators for subsequent analysis. Data collected from four independent biological replicates. (**C**) Survival fraction of cells sorted through the untreated control, low, and high tachyplesin-NBD accumulator gates presented in A and B. Bars and error bars represent the mean and standard deviation of data obtained from four biological replicates, each comprising three technical replicates. Statistical significance was assessed using an ordinary one-way ANOVA with Tukey's multiple comparisons test and confidence level at 95%. ****p<0.0001, ns: p>0.05. (**D–F**) Representative microscopic images depicting brightfield (**D**), tachyplesin-NBD fluorescence (**E**), and propidium iodide (PI) fluorescence (**F**) after exposure to 46 µg mL$^{-1}$ (18.2 µM) tachyplesin-NBD in M9 for 60 min, followed by a M9 wash for 60 min, then 30 µM PI staining for 15 min at 37 °C in the microfluidic mother machine. Blue stars and red diamonds indicate low and high accumulators, respectively. Scale bar indicates 2.5 µm. The left part of (**E**) is reproduced from part of *Figure 1G*. (**G**) Correlation between tachyplesin-NBD and PI fluorescence of N=371 individual *E. coli* cells collected from three independent biological replicates. The purple dashed line shows a nonlinear regression (semi-log) of the data (r$^2$=0.70). (**H**) Correlation between the proportion of low accumulators as a percentage of the whole bacterial population measured via flow cytometry and survival fraction measured via colony-forming unit assays. Symbols and error bars represent the mean and standard deviation of three independent biological replicates, each comprising three technical replicates. Tachyplesin-NBD treatment concentration indicated by colour gradient. Some error bars are masked by the data points. The purple dashed line illustrates a linear regression of the data (r$^2$=0.98, p-value <0.0001).

The online version of this article includes the following source data and figure supplement(s) for figure 2:

**Source data 1.** FACS, colony-forming unit, and microscopy data were used to generate the graphs presented in *Figure 2*.

**Figure supplement 1.** Fluorescence-activated cell sorting of untreated cells, low and high tachyplesin-nitrobenzoxadiazole (NBD) accumulators.

Our analysis revealed significant differential expression of 1,563 genes in at least one of the three groups compared to untreated cells (*Figure 3—source data 2*). By performing cluster analysis of these genes (*Glover et al., 2022*; *Goode et al., 2021a*) (see Methods), we identified five distinct clusters of genes based on their expression profile across the three groups relative to the control population (*Figure 3A, B*, *Figure 3—figure supplement 2*). Gene ontology enrichment revealed significant enrichment of biological processes in clusters 2, 4, and 5 (*Figure 3C and D*). Genes in clusters 2 and 5 were downregulated to a greater extent in low accumulators; genes in cluster 4 were upregulated to a greater extent in low accumulators. This analysis enabled the identification of major transcriptional differences between low and high accumulators of tachyplesin.

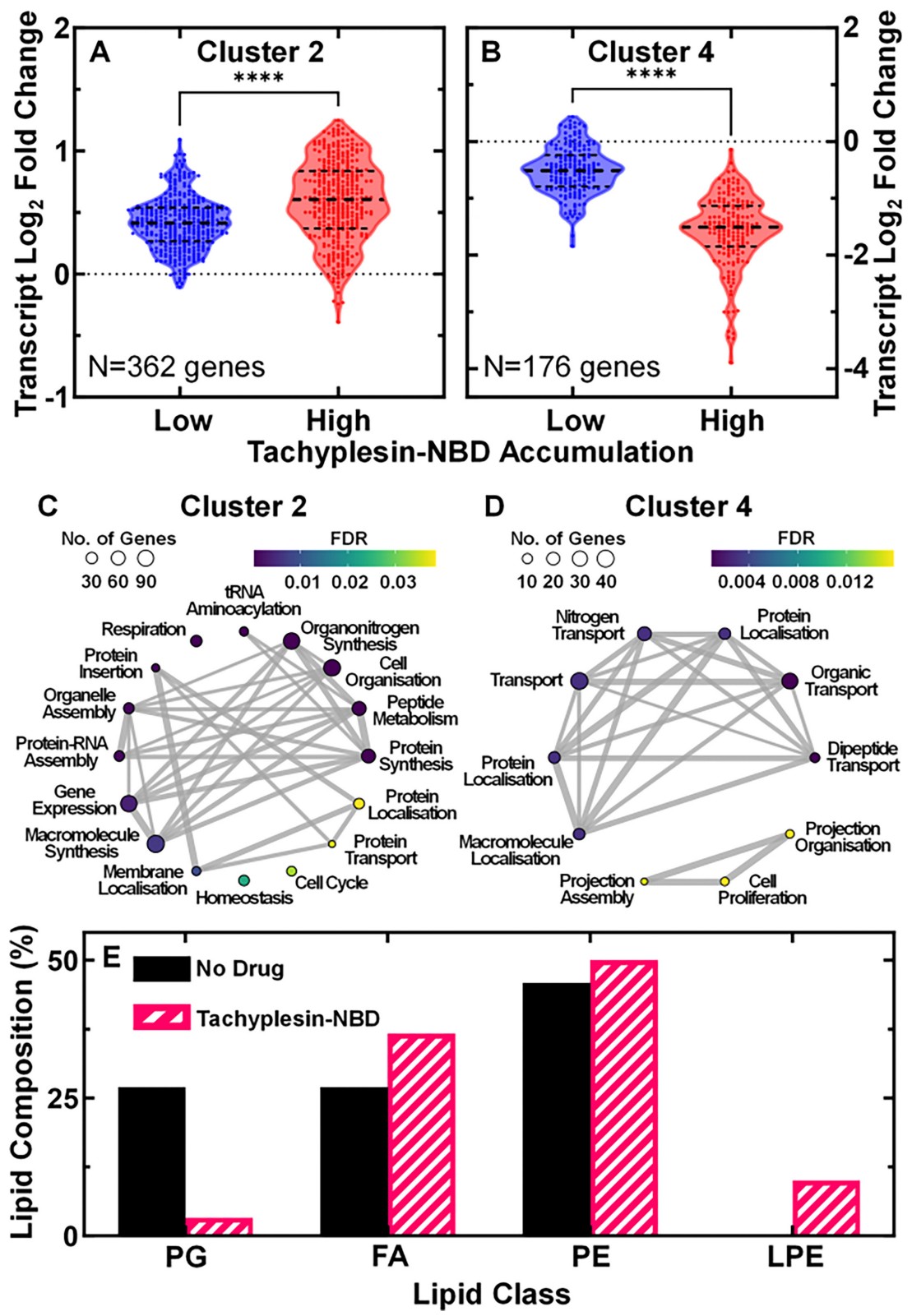

**Figure 3.** Biological processes associated with low tachyplesin accumulation. (**A, B**) Log$_2$ fold changes in transcript reads of genes in low and high tachyplesin-nitrobenzoxadiazole (NBD) accumulators (blue and red violin plots, respectively) relative to untreated stationary phase *E. coli* bacteria in cluster 2 (**A**) and cluster 4 (**B**). Each dot represents a single gene, dashed lines indicate the median and quartiles of each distribution. Dotted lines represent a log$_2$ fold change of 0. Statistical significance was tested using a paired two-tailed Wilcoxon nonparametric test (due to non-normally

*Figure 3 continued on next page*

*Figure 3 continued*

distributed data) with a two-tailed p-value and confidence level at 95%. ****p-value <0.0001. The full list of genes belonging to each cluster is reported in *Figure 3—source data 2* and the violin plots of log₂ fold changes for all clusters are shown in *Figure 3—figure supplement 2*. Data were collected from four independent biological replicates, then pooled for analysis. (**C, D**) Corresponding gene ontology enrichment plots of biological processes enriched in cluster 2 (**C**) and 4 (**D**). Each dot represents a biological process with its size indicating the number of genes associated with each process and the colour indicating the false discovery rate (FDR). The lines and their thickness represent the abundance of mutual genes present within the connected biological processes. The enrichment plot for cluster 5 is not shown as only one biological process, 'response to biotic stimulus,' was enriched. Full details about each process are reported in *Figure 3—source data 3*. (**E**) Relative abundance of lipid classes (PG, phosphatidylglycerol; FA, fatty acids; PE, phosphatidylethanolamine; LPE, lysophosphatidylethanolamine) in untreated bacteria and bacteria treated with 46 µg mL⁻¹ (18.2 µM) tachyplesin-NBD for 60 min.

The online version of this article includes the following source data and figure supplement(s) for figure 3:

**Source data 1.** Transcriptomic and lipidomic data used to generate the graphs presented in *Figure 3*.

**Source data 2.** Differential gene expression in low, 1:1 mix, and high tachyplesin-nitrobenzoxadiazole (NBD) accumulators.

**Source data 3.** Biological processes are significantly enriched in each transcriptomic cluster.

**Source data 4.** Differential transcription factor activity between low and high tachyplesin-nitrobenzoxadiazole (NBD) accumulators.

**Figure supplement 1.** Principal component analysis of transcriptomes.

**Figure supplement 2.** Clustering of differentially regulated genes in low, 1:1 mix, and high accumulators of tachyplesin-nitrobenzoxadiazole (NBD).

**Figure supplement 3.** Clustering of lipidomes of low and high accumulators of tachyplesin-nitrobenzoxadiazole (NBD).

**Figure supplement 4.** Inferred transcription factor (TF) activity for the ten TFs with highest inferred activity in low and high tachyplesin-nitrobenzoxadiazole (NBD) accumulators.

Cluster 2 included processes involved in protein synthesis, energy production, and gene expression that were downregulated to a greater extent in low accumulators than high accumulators (see *Appendix 1—table 1* and *Figure 3—source data 3* for a short and complete list of genes involved in these processes, respectively).

Cluster 4 included processes involved in the transport of organic substances that were upregulated to a greater extent in low accumulators than high accumulators (*Figure 3B and D*). Notably, genes encoding major facilitator superfamily (MFS) and resistance-nodulation-division (RND) efflux pump components, such as *emrK*, *mdtM*, *mdtE*, and *mdtF*, were upregulated in low accumulators (*Appendix 1—table 1* and *Figure 3—source data 3*).

Low and high accumulators displayed differential expression of biological pathways facilitating the synthesis and assembly of the lipopolysaccharides (LPS), which is the first site of interaction between gram-negative bacteria and AMPs (*Kushibiki et al., 2014*). For example, *clsB*, encoding cardiolipin synthase B, was contained in cluster 2 and downregulated in low accumulators (*Appendix 1—table 1* and *Figure 3—source data 3*), suggesting that low accumulators might display a lower content of cardiolipin that could bind tachyplesin due to its negative charge (*Edwards et al., 2017*; *Matsuzaki et al., 1991*). Moreover, we performed a comparative lipidomic analysis of untreated versus tachyplesin-NBD-treated stationary phase *E. coli* using ultra-high performance liquid chromatography-quadrupole time-of-flight mass spectrometry *Casula et al., 2023*. In accordance with our transcriptomic data, we found that treatment with tachyplesin-NBD caused a strong decrease in phosphatidylglycerol (PG from 37% to 3%, i.e. the main constituent of cardiolipin), along with the emergence of lysophosphatidylethanolamines (LPE, 10%) and slight increases in PE and FA (*Figure 3E*, *Figure 3—figure supplement 3*, and *Appendix 1—table 2*).

*ompA*, *ompC*, *degP*, *mcrB*, *tolA*, *tolB*, and *pal* were downregulated in low accumulators, and their deletion has previously been associated with increased secretion of outer membrane vesicles (OMVs) that can confer resistance to AMPs and small molecule antibiotics (*Murray et al., 2020*; *Balhuizen et al., 2021*). Moreover, *nlpA*, *lysS*, *waaC*, and *waaF* were upregulated in low accumulators, and their overexpression has been previously associated with enhanced OMV secretion (*McBroom and Kuehn, 2007*; *Liu et al., 2016*; *Appendix 1—table 1* and *Figure 3—source data 3*).

Finally, low accumulators displayed an upregulation of peptidases and proteases compared to high accumulators, suggesting a potential mechanism for degrading tachyplesin (*Appendix 1—table 1* and *Figure 3—source data 3*).

Noteworthy, tachyplesin-NBD has antibiotic efficacy (see *Figure 2*) and has an impact on the *E. coli* transcriptome (*Figure 3*). Therefore, we cannot conclude whether the transcriptomic differences

reported above between low and high accumulators of tachyplesin-NBD are causative for the distinct accumulation patterns or if they are a consequence of differential accumulation and downstream phenotypic effects.

Next, we sought to infer transcription factor (TF) activities via differential expression of their known regulatory targets (*Alvarez et al., 2016*). A total of 126 TFs were inferred to exhibit differential activity between low and high accumulators (*Figure 3—source data 4* and *Lee et al., 2025*). Among the top ten TFs displaying higher inferred activity in low accumulators compared to high accumulators, four regulate transport systems, i.e., Nac, EvgA, Cra, and NtrC (*Figure 3—figure supplement 4*). However, further experiments should be carried out to directly measure the activity of these TFs.

Taken together, these data suggest that phenotypic variants within clonal bacterial populations display a differential regulation of lipid composition, efflux, outer membrane vesicle secretion, and proteolytic processes that affect tachyplesin accumulation levels.

## Low accumulators display enhanced efflux activity

Next, we set out to delve deeper into the specific molecular mechanisms utilised by low accumulators in response to tachyplesin exposure. We investigated the accumulation kinetics of tachyplesin-NBD in individual stationary phase *E. coli* cells after different durations of tachyplesin-NBD treatment. Strikingly, within just 15 min of treatment, we observed high levels of tachyplesin-NBD accumulation in all cells, placing the whole population in the high accumulator group with a median fluorescence of 15,000 a.u. (*Figure 4A*). However, after 30 min of treatment, a subpopulation began to display lower fluorescence, and after 60 min, the low accumulator fluorescence distribution became evident with a median fluorescence of 3600 a.u. (*Figure 4A*). By contrast, the median fluorescence of high accumulators increased from 15,000 a.u. at 15 min to 32,000 a.u. after 120 min (*Figure 4A*). Taken together, these data demonstrate that the entire isogenic *E. coli* population initially accumulates tachyplesin-NBD to high levels, but a subpopulation can reduce intracellular accumulation of the drug in response to treatment while the other subpopulation continues to accumulate the drug to higher levels. Reduced intracellular accumulation of tachyplesin-NBD in the presence of extracellular tachyplesin-NBD could be due to decreased drug influx, increased drug efflux, increased proteolytic activity, or increased secretion of OMVs.

Next, we performed efflux assays using ethidium bromide (EtBr) by adapting a previously described protocol (*Whittle et al., 2019*). Briefly, we preloaded stationary phase *E. coli* with EtBr by incubating cells at a concentration of 254 µM EtBr in LB for 90 min. Cells were then pelleted and resuspended in M9 to remove extracellular EtBr. Single-cell EtBr fluorescence was measured at regular time points in the absence of extracellular EtBr using flow cytometry. This analysis revealed a progressive homogeneous decrease of EtBr fluorescence due to efflux from all cells within the stationary phase *E. coli* population (*Figure 4—figure supplement 1*). In contrast, when we performed efflux assays by preloading cells with tachyplesin-NBD (46 µg mL$^{-1}$ or 18.2 µM), followed by pelleting and resuspension in M9 to remove extracellular tachyplesin-NBD, we observed a heterogeneous decrease in tachyplesin-NBD fluorescence in the absence of extracellular tachyplesin-NBD: a subpopulation retained high tachyplesin-NBD fluorescence, i.e., high accumulators; whereas another subpopulation displayed decreased tachyplesin-NBD fluorescence, 60 min after the removal of extracellular tachyplesin-NBD (*Figure 4B*). Since these assays were performed in the absence of extracellular tachyplesin-NBD, decreased tachyplesin-NBD fluorescence could not be ascribed to decreased drug influx or increased secretion of OMVs in low accumulators, but could be due to either enhanced efflux or proteolytic activity in low accumulators.

Next, we repeated efflux assays using EtBr in the presence of 46 µg mL$^{-1}$ (or 20.3 µM) extracellular tachyplesin-1. We observed a heterogeneous decrease in EtBr fluorescence with a subpopulation retaining high EtBr fluorescence (i.e. high tachyplesin accumulators) and another population displaying reduced EtBr fluorescence (i.e. low tachyplesin accumulators, *Figure 4—figure supplement 1*) when extracellular tachyplesin-1 was present. Moreover, we repeated tachyplesin-NBD efflux assays in the presence of M9 containing 50 µg mL$^{-1}$ (244 µM) carbonyl cyanide m-chlorophenyl hydrazone (CCCP), an ionophore that disrupts the proton motive force (PMF) and is commonly employed to abolish efflux and found that all cells retained tachyplesin-NBD fluorescence (*Figure 4—figure supplement 2*). However, it is important to note that CCCP does not only abolish efflux but also other respiration-associated and energy-driven processes (*Sharma et al., 2019*).

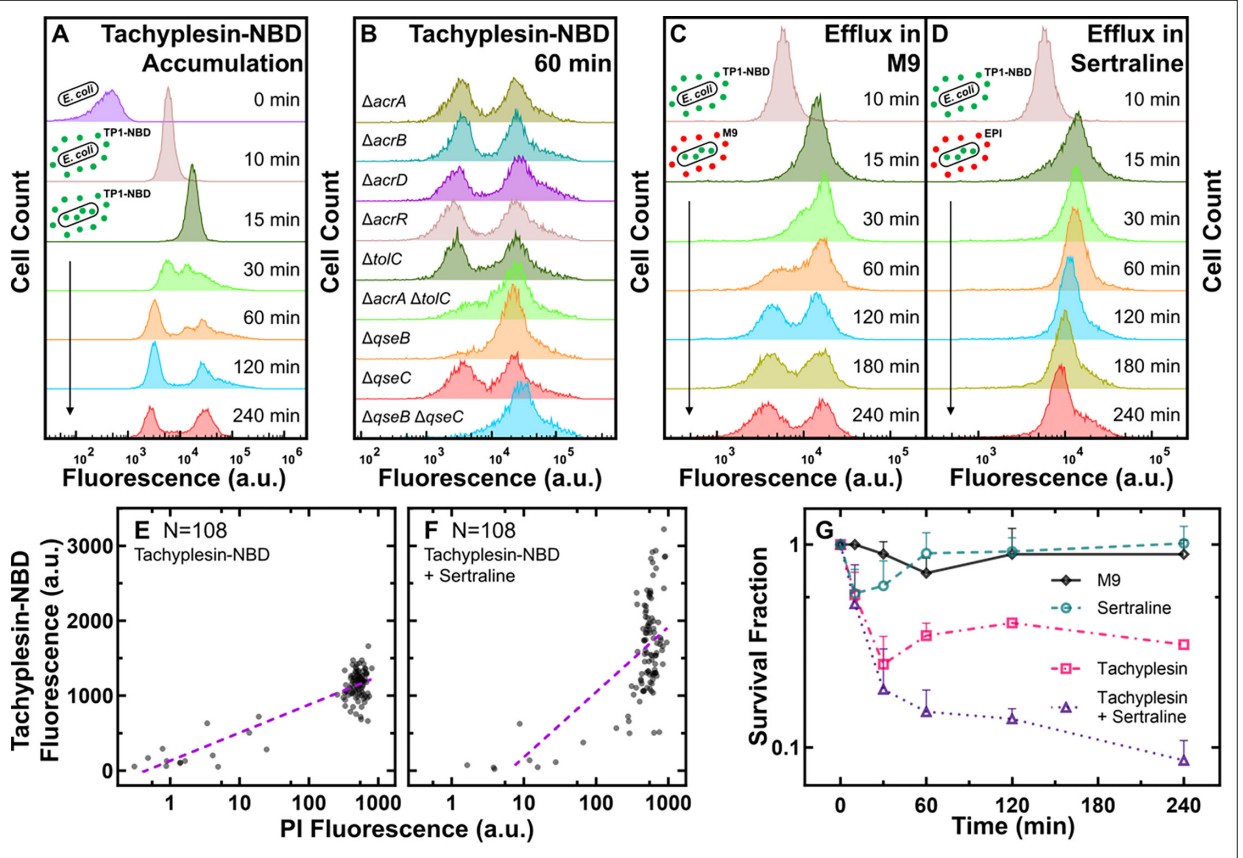

**Figure 4.** Low accumulators of tachyplesin-NBD display enhanced efflux activity. (**A**) Distribution of tachyplesin-nitrobenzoxadiazole (NBD) accumulation in stationary phase *E. coli* BW25113 over 240 min of treatment with 46 µg mL⁻¹ (18.2 µM) tachyplesin-NBD in M9 at 37 °C. (**B**) Tachyplesin-NBD accumulation in *E. coli* BW25113 single or double gene-deletion mutants lacking efflux components or regulators of efflux components. Stationary phase populations of each mutant were treated with 46 µg mL⁻¹ (18.2 µM) tachyplesin-NBD at 37 °C for 60 min and single-cell fluorescence was measured via flow cytometry. (**C, D**) Efflux of tachyplesin-NBD over 240 min, after an initial 15 min preloading with 46 µg mL⁻¹ (18.2 µM) tachyplesin-NBD in M9 then washed and transferred into either M9 (**C**), or sertraline (30 µg mL⁻¹ or 98 µM) (**D**). The top histograms of panels B-D are reproduced from the 10 min histogram of panel A. Histograms in panels A-D are representative of three independent biological replicates and report 10,000 events. (**E, F**) Correlation of tachyplesin-NBD and PI fluorescence of individual stationary phase *E. coli* BW25113 cells measured in the microfluidic mother machine after 60 min of 46 µg mL⁻¹ (18.2 µM) tachyplesin-NBD treatment (N=108) (**E**), or 46 µg mL⁻¹ (18.2 µM) tachyplesin-NBD cotreatment with 30 µg mL⁻¹ (98 µM) sertraline (N=108) (**F**). Purple dashed lines show nonlinear (semi-log) regressions (r²=0.74 and 0.38, respectively). (**G**) Survival fraction of stationary phase *E. coli* BW25113 over 240 min treatment with tachyplesin-1 (46 µg mL⁻¹ or 20.3 µM, magenta squares), sertraline (30 µg mL⁻¹ or 98 µM, green circles), tachyplesin-1 (46 µg mL⁻¹ or 20.3 µM) cotreatment with sertraline (30 µg mL⁻¹ or 98 µM) (purple triangles), or incubation in M9 (black diamonds). Symbols and error bars indicate the mean and standard deviation of measurements performed in three independent biological replicates comprising three technical replicates each. Some error bars are masked behind the symbols.

The online version of this article includes the following source data and figure supplement(s) for figure 4:

**Source data 1.** Flow cytometry, microscopy, and colony-forming unit data used to generate the graphs presented in *Figure 4*.

**Figure supplement 1.** Ethidium bromide efflux in the presence and absence of unlabelled tachyplesin-1.

**Figure supplement 2.** Impact of efflux component and transcription factor deletion on tachyplesin-nitrobenzoxadiazole (NBD) accumulation.

**Figure supplement 3.** Tachyplesin-nitrobenzoxadiazole (NBD) efflux in the presence of different efflux pump inhibitors.

**Figure supplement 4.** Differences in metabolic activity of low and high tachyplesin-nitrobenzoxadiazole (NBD) accumulators.

Taken together, our data demonstrate that in the absence of extracellular tachyplesin, stationary phase *E. coli* homogeneously efflux EtBr, whereas only low accumulators are capable of performing efflux of intracellular tachyplesin after initial tachyplesin accumulation. In the presence of extracellular tachyplesin, only low accumulators can perform efflux of both intracellular tachyplesin and intracellular EtBr. However, it is also conceivable that besides enhanced efflux, low accumulators employ proteolytic activity, OMV secretion, and variations to their bacterial membrane to hinder further uptake and intracellular accumulation of tachyplesin in the presence of extracellular tachyplesin.

Next, we performed tachyplesin-NBD accumulation assays using 28 single and four double *E. coli* BW25113 gene-deletion mutants of efflux components and transcription factors regulating efflux. While for the majority of the mutants we recorded bimodal distributions of tachyplesin-NBD accumulation similar to the distribution recorded for the *E. coli* BW25113 parental strain (**Figure 4**, **Figure 4— figure supplement 2**), we found unimodal distributions of tachyplesin-NBD accumulation constituted only of high accumulators for both Δ*qseB* and Δ*qseB*Δ*qseC* mutants as well as reduced numbers of low accumulators for the Δ*acrA*Δ*tolC* mutant (**Figure 4B**). Considering that the AcrAB-TolC tripartite RND efflux system is known to confer genetic resistance against AMPs like protamine and polymyxin-B (**Weatherspoon-Griffin et al., 2014**; **Lister et al., 2012**) and that the quorum sensing regulators *qseBC* might control the expression of *acrA* (**Li et al., 2020**), these data further corroborate the hypothesis that low accumulators can efflux tachyplesin and survive treatment with this AMP.

## Tachyplesin accumulation and efficacy can be boosted with the addition of efflux pump inhibitors or nutrients

Next, we screened a panel of seven alternative efflux pump inhibitors (EPIs) to identify compounds capable of preventing the formation of low accumulators of tachyplesin-NBD. Interestingly, M9 containing 30 µg mL$^{-1}$ (98 µM) sertraline (**Figure 4D**, **Figure 4—figure supplement 2**), an antidepressant which inhibits efflux activity of RND pumps, potentially through direct binding to efflux pumps (**Bohnert et al., 2011**) and decreasing the PMF (**Li et al., 2017**), or 50 µg mL$^{-1}$ (110 µM) verapamil (**Figure 4—figure supplement 3**), a calcium channel blocker that inhibits MATE transporters **Radchenko et al., 2015** by a generally accepted mechanism of PMF generation interference (**Rodrigues et al., 2012**; **Mahamoud et al., 2007**), was able to prevent the emergence of low accumulators. Furthermore, tachyplesin-NBD cotreatment with sertraline simultaneously increased tachyplesin-NBD accumulation levels in individual cells and the proportion of PI-stained cells (**Figure 4E and F**). The use of berberine, a natural isoquinoline alkaloid that inhibits MFS transporters (**Ying and Xizhen, 2023**) and RND pumps (**Morita et al., 2016**), potentially by inhibiting conformational changes required for efflux activity (**Ying and Xizhen, 2023**), and baicalein, a natural flavonoid compound that inhibits ABC (**Wang et al., 2019**) and MFS (**Fujita et al., 2005**; **Chan et al., 2011**) transporters, potentially through PMF dissipation (**Wang et al., 2023**), led to the formation of a unimodal distribution of low accumulators of tachyplesin (**Figure 4—figure supplement 3**). Phenylalanine-arginine beta-naphthylamide (PAβN), a synthetic peptidomimetic compound that inhibits RND pumps (**Cortez-Cordova and Kumar, 2011**) through competitive inhibition (**Compagne et al., 2023**), reserpine, an indole alkaloid that inhibits ABC and MFS transporters, and RND pumps (**Shaheen et al., 2019**), by altering the generation of the PMF (**Mahamoud et al., 2007**), and 1-(1-naphthylmethyl)piperazine (NMP), a synthetic piperazine derivative that inhibits RND pumps (**Bohnert and Kern, 2005**), through non-competitive inhibition (**Vargiu et al., 2014**), did not prevent the emergence of low accumulators (**Figure 4—figure supplement 3**).

Next, we investigated whether sertraline could enhance the bactericidal activity of the parent drug, tachyplesin-1. Using colony-forming unit assays, we measured the survival fraction of stationary phase *E. coli* treated with either tachyplesin-1 at 46 µg mL$^{-1}$ (20.3 µM) (which has an MIC value of 1 µg mL$^{-1}$ or 0.43 µM against *E. coli* **Łapińska et al., 2022**) or sertraline at 30 µg mL$^{-1}$ (98 µM) (for which we determined an MIC value of 128 µg mL$^{-1}$ or 418 µM against *E. coli*, see **Figure 4—source data 1**), or a combination of both compounds, and compared these data to untreated *E. coli* incubated in M9. The survival fraction after treatment with sertraline was comparable to that measured for untreated *E. coli*, whereas the survival fraction measured after the combination treatment was fivefold lower than that measured after tachyplesin-1 treatment (**Figure 4G**). Increased tachyplesin efficacy in the presence of sertraline is likely due to efflux inhibition. However, it is also conceivable that increased tachyplesin efficacy is due to metabolic and transcriptomic changes induced by sertraline.

Our transcriptomics analysis also showed that low tachyplesin accumulators downregulated genes associated with protein synthesis, energy production, and gene expression compared to high accumulators. To gain further insight into the metabolic state of low tachyplesin accumulators, we employed the membrane-permeable redox-sensitive dye, resazurin, which is reduced to the highly fluorescent resorufin in metabolically active cells. We first treated stationary phase *E. coli* with 46 µg mL$^{-1}$ (18.2 µM) tachyplesin-NBD for 60 min, then washed the cells, and then incubated them in 1 µM resazurin and 50 µM CCCP for 15 min and measured single-cell fluorescence of resorufin and

tachyplesin-NBD simultaneously via flow cytometry. CCCP was included to minimise differences in efflux activity and preserve resorufin retention between low and high accumulators, though some variability in efflux may still persist. We found that low tachyplesin-NBD accumulators also displayed low fluorescence of resorufin, whereas high tachyplesin-NBD accumulators also displayed high fluorescence of resorufin (*Figure 4—figure supplement 4*), suggesting lower metabolic activity in low tachyplesin-NBD accumulators.

Therefore, we set out to test the hypothesis that the formation of low accumulators could be prevented by manipulating the nutrient environment around the bacteria. We treated stationary phase *E. coli* with 46 µg mL⁻¹ (18.2 µM) tachyplesin-NBD either in M9, or M9 supplemented with 0.4% glucose and 0.2% casamino acids. After 15 min of treatment, both conditions resulted in a unimodal fluorescence distribution, with median values of 15,000 a.u. and 18,000 a.u., respectively (*Figures 4A and 5A*, respectively). However, upon extending the treatment to 30 min, we observed distinct responses to tachyplesin-NBD treatment. For *E. coli* treated with tachyplesin-NBD in M9, the emergence of low accumulators led to a bimodal distribution of single-cell fluorescence (*Figure 4A*). In contrast, when *E. coli* was treated with tachyplesin-NBD in M9 supplemented with nutrients, the fluorescence of the entire population increased, resulting in a unimodal distribution of single-cell fluorescence with a median of 35,000 a.u. (*Figure 5A*).

Similarly, *E. coli* incubated in LB for 3 h before drug treatment, and therefore in the exponential phase of growth, displayed a unimodal fluorescence distribution with a median of 80,000 a.u. after 10 min treatment in tachyplesin-NBD in M9 (*Figure 5B*). These exponential phase bacteria also displayed a modest increase in cell size compared to stationary phase *E. coli* (*Figure 5—figure supplement 1*). Interestingly, a small subpopulation of low accumulators emerged within exponential phase *E. coli* after 120 min tachyplesin-NBD treatment in M9, with a median of 4700 a.u. (*Figure 5B*). This subpopulation contributed to 10% of the entire isogenic *E. coli* population, five-fold less abundant compared to stationary phase *E. coli* treated in tachyplesin-NBD in M9 (*Figure 4A*). Finally, we investigated whether preventing the formation of low accumulators via nutrient supplementation could enhance the efficacy of tachyplesin treatment by performing colony-forming unit assays. We observed that tachyplesin exhibited enhanced efficacy in eradicating *E. coli* both when M9 was supplemented with nutrients, and against exponential phase *E. coli* compared to stationary phase *E. coli* without nutrient supplementation (*Figure 5C*).

## Discussion

The link between antimicrobial accumulation and efficacy has not been widely investigated, especially in the context of AMPs (*Rybenkov et al., 2021*). In fact, it has been widely accepted that the cell membrane is the primary target of most AMPs and that their main mechanism of action is membrane disruption. However, emerging evidence suggests that AMPs may also enter the intracellular environment and target intracellular processes, creating opportunities for the development of bacterial resistance (*Rima et al., 2021*; *Mookherjee et al., 2020*; *Magana et al., 2020*; *Bechinger and Gorr, 2017*).

Here, we demonstrate for the first time that there is a strong correlation between tachyplesin intracellular accumulation and efficacy. We show that after an initial homogeneous tachyplesin accumulation within a stationary phase *E. coli* population, tachyplesin is retained intracellularly by bacteria that do not survive tachyplesin exposure, whereas tachyplesin is retained only in the membrane of bacteria that survive tachyplesin exposure.

These findings align with evidence showing that modifications to tachyplesin that preserve membrane disruption but reduce membrane translocation, such as linear analogues or PEGylation, result in diminished antimicrobial activity (*Imura et al., 2007*; *Matsuzaki et al., 1997*). Furthermore, tachyplesin has also been reported to interact with intracellular targets such as the minor groove of DNA (*Imura et al., 2007*; *Yonezawa et al., 1992*), intracellular esterases (*Hong et al., 2015*), or the 3-ketoacyl carrier protein reductase FabG (*Liu et al., 2018*). However, our transcriptomic analysis did not reveal differential regulation of any of these pathways between low and high accumulators of tachyplesin, suggesting that bacteria likely employ alternative mechanisms rather than target downregulation to survive tachyplesin treatment, a point on which we expand below.

The emergence of genetic resistance to AMPs is strikingly lower compared to resistance to small molecule antibiotics (*Spohn et al., 2019*; *Bechinger and Gorr, 2017*; *Kintses et al., 2019a*). However, research on AMP resistance has often focused on whole population responses (*Spohn et al., 2019*;

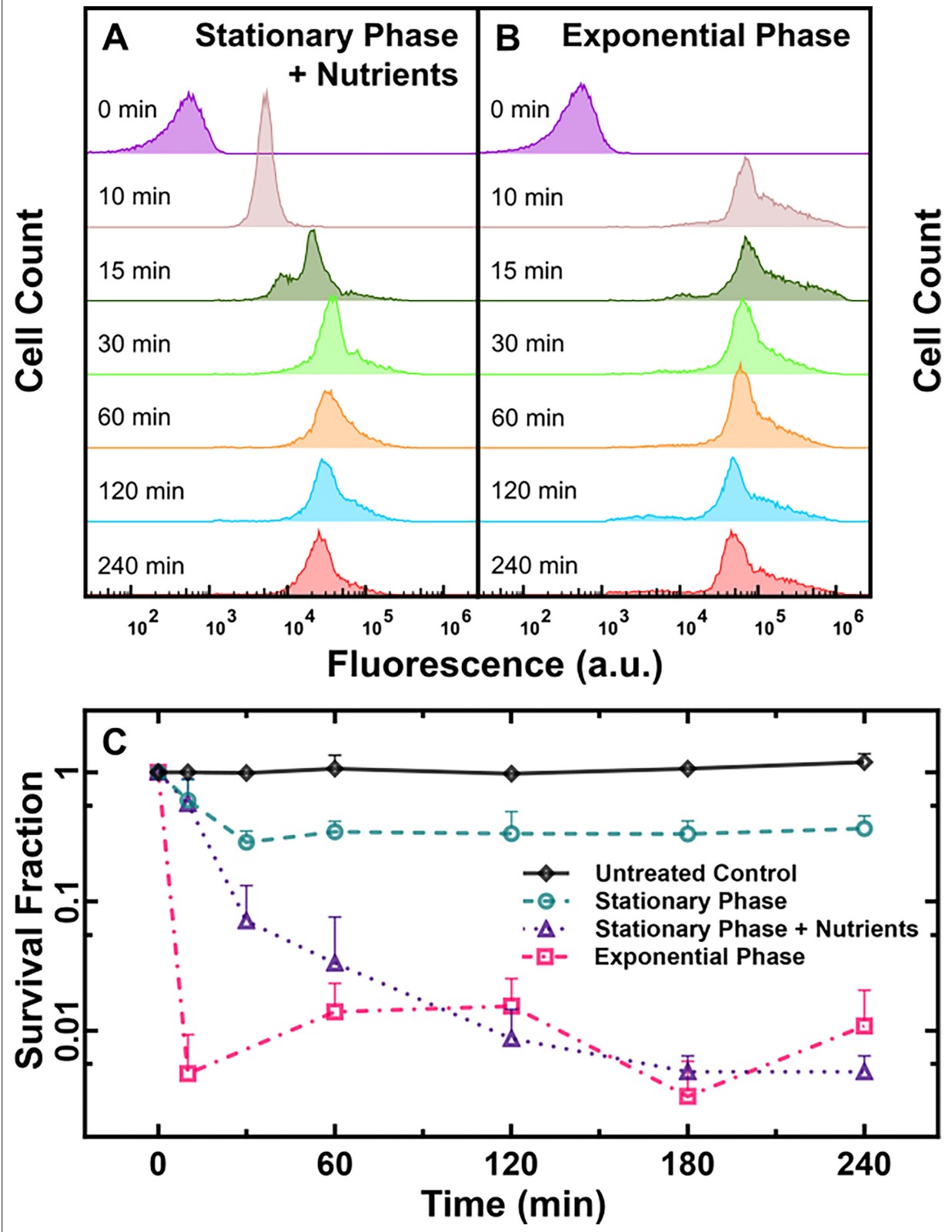

**Figure 5.** Impact of nutritional environment and growth phase on tachyplesin-nitrobenzoxadiazole (NBD) accumulation and efficacy. (**A, B**) Temporal dependence of the distribution of tachyplesin-NBD accumulation in stationary phase *E. coli* in M9 supplemented with 0.4% glucose and 0.2% casamino acids (**A**) and exponential phase *E. coli* in M9 only (**B**) treated with 46 µg mL⁻¹ (18.2 µM) tachyplesin-NBD. Each distribution is representative of three independent biological replicates. (**C**) Survival fraction of untreated stationary phase *E. coli* in M9 only (black diamonds), and bacteria treated with 46 µg

*Figure 5 continued on next page*

*Figure 5 continued*

mL⁻¹ (18.2 µM) tachyplesin-NBD over 240 min in stationary phase *E. coli* in M9 only (green circles), stationary phase *E. coli* treated in M9 supplemented with 0.4% glucose and 0.2% casamino acids (purple triangles), exponential phase *E. coli* treated in M9 only (magenta squares). Symbols and error bars indicate the mean and standard deviation of three biological replicates measured from three technical replicates each. Some error bars are masked behind the symbols.

The online version of this article includes the following source data and figure supplement(s) for figure 5:

**Source data 1.** Flow cytometry and colony-forming unit data were used to generate the graphs presented in *Figure 5*.

**Figure supplement 1.** Impact of the bacterial phase of growth on cell size.

*Bechinger and Gorr, 2017*; *Kintses et al., 2019a*; *Hong et al., 2022*), overlooking heterogeneous responses to drug treatment in isogenic bacteria. Crucially, phenotypic variants can survive treatment with small molecule antibiotics through several distinct phenomena, including persistence (*Balaban et al., 2019*), the viable but nonculturable state (*Ayrapetyan et al., 2018*), heteroresistance (*Andersson et al., 2019*), tolerance (*Windels et al., 2019b*), and perseverance (*Brandis et al., 2023*).

Here, we report the first observation of phenotypic resistance to tachyplesin within isogenic *E. coli* and *P. aeruginosa* populations. This novel phenomenon is characterised by a bimodal distribution of tachyplesin accumulation, revealing two distinct subpopulations: low (surviving) and high (susceptible) accumulators. This bimodal distribution draws similarities to the perseverance phenomenon identified for two antimicrobials possessing intracellular targets, rifampicin and nitrofurantoin. Perseverance is characterised by a bimodal distribution of single-cell growth rates, where the slower-growing subpopulation maintains growth during drug treatment (*Brandis et al., 2023*).

Bacteria can reduce the intracellular concentration of antimicrobials via active efflux, allowing them to survive in the presence of high extracellular antimicrobial concentrations (*Blair et al., 2015*). Efflux as a mechanism of tachyplesin resistance has been shown at the population level in tachyplesin-resistant *P. aeruginosa* via transcriptome analysis (*Hong et al., 2022*; *Hong et al., 2020*), whereas phenotypic resistance to tachyplesin has not been investigated. Heterogeneous expression of efflux pumps within isogenic bacterial populations has been reported (*Pu et al., 2016*; *El Meouche and Dunlop, 2018*; *Łapińska et al., 2022*; *Bergmiller et al., 2017*; *El Meouche et al., 2016*; *Le et al., 2021*). However, recent reports have suggested that efflux is not the primary mechanism of antimicrobial resistance within stationary phase bacteria (*Whittle et al., 2021*; *Mitchell et al., 2017*) and the impact of heterogeneous efflux pump expression on AMP accumulation is unknown. Here, we show for the first time the upregulation of a range of MFS, ABC, and RND efflux pumps in a stationary phase subpopulation that exhibited low accumulation of tachyplesin with corresponding enhanced survival and that the deletion of two components of the AcrAB-TolC efflux pump or the quorum sensing regulator QseBC leads to a dramatic decrease in the numbers of low accumulators.

Moreover, whether certain resistance mechanisms utilised by phenotypic variants surviving antimicrobial treatment are preemptive or induced by drug treatment is underinvestigated (*Balaban et al., 2019*; *Fernández and Hancock, 2012*) primarily due to the requirement of prior selection for phenotypic variants (*Pu et al., 2016*). As our accumulation assay did not require the prior selection for phenotypic variants, we have demonstrated that low accumulators emerge subsequent to the initial high accumulation of tachyplesin-NBD, suggesting enhanced efflux as an induced response. However, it is conceivable that other pre-existing traits of low accumulators also contribute to reduced tachyplesin accumulation. For example, reduced protein synthesis, energy production, and gene expression in low accumulators could slow down tachyplesin efficacy, giving low accumulators more time to mount efflux as an additional protective response.

Furthermore, most AMPs are cationic and interact electrostatically with the negatively charged LPS in the bacterial outer membrane (*Lei et al., 2019*). To reduce affinity to AMPs, bacteria can modify their membrane charge, thickness, fluidity, or curvature (*MacDermott-Opeskin et al., 2022*). At the population level, bacteria have been reported to add positively charged L-Ara4N or phosphoethanolamine (pEtN) to LPS lipid-A to resist AMPs (*Bechinger and Gorr, 2017*). Accordingly, our transcriptomics analysis revealed the upregulation in the low accumulator subpopulation of the *arn* operon that is involved in the biosynthesis and attachment of the positively charged L-Ara4N to lipid-A (*Yan et al., 2007*). Furthermore, tachyplesin has been shown to bind strongly to negatively charged PG lipids (*Matsuzaki et al., 1991*). Accordingly, our lipidomics data revealed that tachyplesin-treated bacteria were composed of significantly lower abundance of PG lipids compared to the untreated bacteria.

OMVs, nano-sized proteoliposomes secreted by Gram-negative bacteria, are a less studied system that allows bacteria to remove AMPs from the cell (*Kulp and Kuehn, 2010*). Increased secretion of OMVs has also been shown to be induced by AMP exposure (*Balhuizen et al., 2021*). OMVs are suggested to counteract AMPs via at least three mechanisms: the adsorption of extracellular AMPs to decrease AMP concentrations in the environment *Murray et al., 2020*; *Balhuizen et al., 2021*; removal of parts of the bacterial outer membrane affected by AMPs *Balhuizen et al., 2021*; and export of antimicrobials from the intracellular compartment (*Medvedeva et al., 2014*). Our transcriptomic data revealed the downregulation in low accumulators of genes, such as *ompA* and *ompC*, known to increase the secretion of OMVs upon deletion, and the upregulation of genes, such as *nlpA*, whose overexpression has been linked to enhanced OMV secretion (*Nevermann et al., 2019*; *MacNair and Tan, 2023*).

Bacteria also employ extracellular or intracellular proteases *Blair et al., 2022*; *Bechinger and Gorr, 2017* and peptidases *Li et al., 2018*; *Mattiuzzo et al., 2014* to cleave and neutralise the activity of AMPs, including tachyplesin (*Hong et al., 2016*). Our transcriptomic analysis revealed the simultaneous upregulation in low accumulators of proteases and peptidases that could potentially degrade tachyplesin either extracellularly or intracellularly.

Responses to extrinsic environmental cues, such as the starvation-induced stringent response, have been shown to induce failure in antimicrobial treatment (*Podlesek and Žgur Bertok, 2020*). These responses are typically mediated by two-component regulatory systems (TCSs) sensing external stimuli via sensor kinases and altering gene expression via response regulators (*Hirakawa et al., 2020*). Additionally, nucleoid-associated proteins (NAPs) influence bacterial chromosome organisation and gene transcription in response to intracellular or extracellular stress factors (*Hołówka and Zakrzewska-Czerwińska, 2020*). Finally, transcriptional regulators such as regulators of the LysR family and histone-like proteins (H-NS) also modulate gene expression (*Wójcicki et al., 2021*). One such TCS, QseBC, was previously inferred to mediate resistance to a tachyplesin analogue by upregulating efflux genes based on transcriptomic analysis and hypersusceptibility of ΔqseBΔqseC mutants (*Yu et al., 2020*). Accordingly, we demonstrated that ΔqseBΔqseC mutants display only high accumulators of tachyplesin-NBD, suggesting that this TCS might play an important role in the control of tachyplesin efflux, possibly by regulating the expression of AcrA and TolC or other efflux components (*Li et al., 2020*). This data provides experimental support for the involvement of QseBC-mediated efflux of tachyplesin, extending previous transcription-based inferences. However, it is also conceivable that the deletion of QseBC has pleiotropic effects on other cellular mechanisms involved in tachyplesin accumulation. We also acknowledge that our data do not take into account post-transcriptional modifications that represent a second control point to survive external stressors.

Drugs that inhibit the efflux activity of bacterial pathogens have been shown to be a promising strategy for repurposing antimicrobials (*Blanco et al., 2018*; *Turner et al., 2019*; *AlMatar et al., 2021*). An FDA-approved drug, sertraline, was found to be effective in preventing the formation of the low accumulator phenotype by inhibiting tachyplesin efflux and enhancing the eradication of stationary phase bacteria, although it is also conceivable that sertraline prevented the formation of the low accumulator phenotype via efflux-independent mechanisms. These data strongly suggest that EPIs are a promising approach for developing new combination therapies against stationary phase bacteria, contrary to previous evidence suggesting their ineffectiveness against stationary phase bacteria (*Whittle et al., 2021*). However, it should be noted that the concentration of sertraline in the plasma of patients using sertraline as an antidepressant is below the concentration that we employed in our study (*Rácz and Spengler, 2023*). This limitation underscores the broader challenge of identifying EPIs that are both effective and minimally toxic within clinically achievable concentrations, while also meeting key therapeutic criteria such as broad-spectrum efficacy against diverse efflux pumps, high specificity for bacterial targets, and non-inducers of AMR (*Zhang et al., 2023c*). However, advances in biochemical, computational, and structural methodologies hold the potential to guide rational drug design, making the search for effective EPIs more promising (*Opperman and Nguyen, 2015*). Therefore, more investigation should be carried out to further optimise the use of sertraline or other EPIs in combination with tachyplesin and other AMPs.

Moreover, growth rate and metabolism have been shown to influence antimicrobial efficacy (*Lopatkin et al., 2019*), with starved or slow growing stationary phase bacteria becoming transiently insensitive to antimicrobials when metabolites or ATP become unavailable (*Tamer et al., 2021*).

For example, the expression of efflux genes has been shown to be increased in *E. coli* in minimal media compared to rich media (*Bailey et al., 2006*). Due to the heightened resistance of starved and stationary phase cells, successful attempts have been made to selectively target metabolically dormant bacteria (*Zheng et al., 2024*) or to sensitise recalcitrant cells via metabolites as a potential therapeutic strategy (*Allison et al., 2011*; *Meylan et al., 2017*; *Gutierrez et al., 2017*). Likewise, we found that stationary phase cells supplemented with glucose and casamino acids were more sensitive to treatment with tachyplesin. However, nutrient supplementation may potentially lead to an increase in bacterial cell count and disease burden, as well as an escalation in selection pressure on surviving cells. Moreover, nutrient supplementation as a therapeutic strategy may not be viable in many infection contexts, as host density and the immune system often regulate access to nutrients (*Tamer and Toprak, 2017*). It may be worthwhile to test other metabolites such as fumarate, which has received FDA approval in several drugs and has been shown to potentiate antimicrobial efficacy (*Meylan et al., 2017*). Therefore, we conclude that the use of EPIs in combination with tachyplesin represents a more viable antimicrobial treatment against antimicrobial-refractory stationary phase bacteria.

## Methods

### Chemicals and cell culture

All chemicals were purchased from Fisher Scientific or Sigma-Aldrich unless otherwise stated. LB medium (10 g $L^{-1}$ tryptone, 5 g $L^{-1}$ yeast extract, and 10 g $L^{-1}$ NaCl) and LB agar plates (LB with 15 g $L^{-1}$ agar) were used for planktonic growth and streak plates, respectively. Tachyplesin-1 and tachyplesin-1-NBD were synthesised by WuXi AppTech (Shanghai, China). Stock solutions of antimicrobials (1.28 mg $mL^{-1}$), PI (1.5 mM), proteinase K (20 mg $mL^{-1}$), and EtBr (1 mg $mL^{-1}$) were obtained by dissolving in Milli-Q water. All EPIs and $C_{12}$-resazurin were obtained by dissolving in dimethyl sulfoxide at a concentration of 1 mg $mL^{-1}$. Carbon-free M9-minimal medium used for dilution of antimicrobials, PI, proteinase K, EtBr, EPIs, $C_{12}$-resazurin, and bacteria was prepared using 5×M9 minimal salts (Merck, Germany), with an additional 2 mM $MgSO_4$ and 0.1 mM $CaCl_2$ in Milli-Q water. 0.4% glucose and 0.2% casamino acids were added to yield nutrient-supplemented M9. M9 was then filtered through a 0.22 µm Minisart Syringe Filter (Sartorius, Germany).

*E. coli* BW25113 and *S. aureus* ATCC 25923 were purchased from Dharmacon (Horizon Discovery, UK). *E. coli* and *K. pneumoniae* (JS1187) clinical isolates were kindly provided by Fernanda Paganelli, UMC Utrecht, the Netherlands. *P. aeruginosa* (PA14 *flgK*::Tn5B30(Tc)) was kindly provided by George O'Toole, Dartmouth College, USA. *E. coli* single-gene deletion KO mutants were obtained from the Keio collection (*Baba et al., 2006*). To construct double gene-deletion mutants, the antibiotic resistance gene flanked by flippase recognition target sequences was removed from the chromosome using the pCP20 plasmid which encodes the flippase enzyme. The resulting transformants were then treated as described previously (*Datsenko and Wanner, 2000*). Double gene-deletion mutants were then subsequently constructed by P1vir transduction of alleles from the Keio collection, as previously detailed (*Thomason et al., 2007*). All strains were stored in a 50% glycerol stock at –80 °C. Streak plates for each strain were produced by thawing a small aliquot of the corresponding glycerol stock every week and plated onto LB agar plates. Stationary phase cultures were prepared by inoculating 100 mL fresh LB medium with a single bacterial colony from a streak plate and incubated on a shaking platform at 200 rpm and 37 °C for 17 h (*Stie et al., 2020*). Exponential phase cultures were prepared by transferring 100 µL of a stationary phase culture into 100 mL LB and incubated on a shaking platform at 200 rpm and 37 °C for 3 h.

### Synthesis of fluorescent antimicrobial derivatives

Fluorescent antimicrobial derivatives of tachyplesin (*Blaskovich et al., 2019*; *Edwards et al., 2017*), arenicin (*Elliott et al., 2020*), polymyxin-B (*Gallardo-Godoy et al., 2016*), and octapeptin *Velkov et al., 2018* were designed and synthesised based on structure-activity-relationship studies and synthetic protocols reported in previous publications (*Zhang et al., 2022*), substituting a non-critical amino acid residue with an azidolysine residue that was then employed for the subsequent Cu-catalysed azide-alkyne cycloaddition 'click' reactions with nitrobenzoxadiazole (NBD)-alkyne. Detailed synthesis and characterisation of these probes will be reported in due course.

## Phylogenetic and AMR genes analyses

The *E. coli* BW25113 genome sequence was downloaded from the National Center for Biotechnology Information's (NCBI) GenBank (*Grenier et al., 2014*). *E. coli* clinical isolates MA02514, MA07534, MA08141, MC04960, and MC42862 were collected and sequenced by UMC Utrecht, the Netherlands (*van den Bunt, 2014*). Briefly, genomic DNA libraries were prepared using the Nextera XT Library Prep Kit (Illumina, USA) and sequenced on either the Illumina MiSeq or NextSeq (Illumina, USA). Contigs were assembled using SPAdes Genome Assembler version 3.6.2 (*Bankevich et al., 2012*), and contigs larger than 500 bp with at least 10× coverage were further analysed. *E. coli* clinical isolates JS1060, JS1073, JS1147, and JS1147 were collected by UMC Utrecht, the Netherlands, and were sequenced by a microbial genome sequencing service (MicrobesNG, UK). Briefly, genomic DNA libraries were prepared using the Nextera XT Library Prep Kit (Illumina, USA) following the manufacturer's protocol with the following modifications: input DNA was increased twofold, and PCR elongation time was increased to 45 s. DNA quantification and library preparation were carried out on a Hamilton Microlab STAR automated liquid handling system (Hamilton Bonaduz AG, Switzerland). Libraries were sequenced on an Illumina NovaSeq 6000 (Illumina, USA) using a 250 bp paired-end protocol. Reads and adapters trimmed using Trimmomatic version 0.30 (*Bolger et al., 2014*) with a sliding window quality cutoff of Q15. De novo assembly was performed on samples using SPAdes version 3.7 (*Bankevich et al., 2012*), and contigs were annotated using Prokka 1.11 (*Seemann, 2014*).

Genomic data for AMR genes were screened through ABRicate (*Seemann, 2016*) and the NCBI's AMRFinderPlus databases (*Feldgarden et al., 2021*). A rooted phylogenetic tree was created from the genomic data by pairwise distance estimation of all genomes and an outgroup genome (NCBI's reference genome for *Escherichia fergusonii*) with the *dist* function from the Mash package (*Ondov et al., 2016*), with a k-mer size of 21. Unrooted phylogenetic trees were created from the resulting pairwise distance matrices using the *nj* (neighbour joining) function in APE (*Paradis et al., 2004*). These trees were then rooted to the outgroup (*ape::root*). The phylogenetic tree was plotted against presence/absence heatmaps of AMR genes using *ggtree* (*Xu et al., 2022*).

## Flow cytometric assays

Bacterial cultures were prepared as described above. For all flow cytometric assays, bacterial cultures (stationary and exponential phase) were adjusted to an $OD_{600}$ of four for *E. coli* and *K. pneumoniae*, five for *S. aureus*, and two for *P. aeruginosa* (*Łapińska et al., 2025*). Cultures were pelleted by centrifugation in a SLS4600 microcentrifuge (Scientific Laboratory Supplies, UK) at 4000 ×*g* and room temperature for 5 min. The supernatant was then removed, and cells were resuspended in the appropriate volume of the treatment solution for incubation, with a final volume between 50 and 550 μL depending on the experiment. All treatments were incubated at 37 °C and 1000 rpm on a ThermoMixer C (Eppendorf, Germany), except for cold treatments, which were incubated on ice, for the stated duration. All wash steps were performed to remove any unaccumulated compounds by centrifugation at 4000 ×*g* for 5 min, after which the supernatant was removed to eliminate any extracellular drug.

For AMP accumulation assays, bacteria were incubated with the stated concentration of peptide in either carbon-free or nutrient-supplemented M9. For proteinase K degradation and $C_{12}$-resazurin metabolic activity assays, *E. coli* was first incubated in 46 μg mL$^{-1}$ (18.2 μM) tachyplesin-NBD for 60 min followed by a wash step. Cells were then incubated in either 20 μg mL$^{-1}$ (0.7 μM) proteinase K, or 1 μM $C_{12}$-resazurin and 50 μM CCCP, all diluted in carbon-free M9. For tachyplesin-NBD efflux assays, *E. coli* was incubated in 46 μg mL$^{-1}$ (18.2 μM) tachyplesin-NBD for 5 min, followed by a wash step and incubation in EPI diluted in carbon-free M9. For EtBr efflux assays, *E. coli* was preloaded with EtBr by adding 100 μg mL$^{-1}$ (254 μM) EtBr to the stationary phase culture 90 min before a total of 17 h incubation (i.e. 15.5 h post-inoculation). Cells were then washed and incubated in 46 μg mL$^{-1}$ (20.3 μM) unlabelled tachyplesin-1 diluted in carbon-free M9. Carbon-free M9 was used as the untreated control in all flow cytometric assays.

At the stated incubation time points, a 30 μL sacrificial aliquot was transferred to microcentrifuge tubes. A wash step was immediately performed, after which cells were resuspended and diluted 500x in carbon-free M9 before flow cytometry measurements. For the accumulation assays, to account for the time between sample collection and flow cytometry measurements, 10 min was added to the

reported incubation time for the first two time points (i.e. the sacrificial aliquot taken immediately after tachyplesin-NBD addition was reported as 10 min).

Flow cytometric measurements were performed on the CytoFLEX S Flow Cytometer (Beckman Coulter, USA) equipped with a 488 nm (50 mW) and a 405 nm (80 mW) laser. Fluorescence of individual bacteria was measured using the fluorescein isothiocyanate (FITC) channel (488 nm 525/40 BP) for fluorescent peptides and phycoerythrin (PE) channel (488 nm 585/42 BP) for EtBr and $C_{12}$-resorufin. Cell size was measured using FSC (forward scatter) and Violet SSC (side scatter) channels. Avalanche photodiode gains of FSC: 1000, SSC: 500, Violet SSC: 1, FITC: 250, and a threshold value of SSC-A: 10,000 to limit background noise was used. Bacteria were gated to separate cells from background noise by plotting FSC-A and Violet SSC-A. Background noise was then further separated based on cellular autofluorescence measured on the FITC-A channel for cells not treated with fluorescent peptides. An additional gate on the FITC-A channel was then used for cells treated with fluorescent peptides to further separate background noise. CytoFLEX Sheath Fluid (Beckman Coulter, USA) was used as sheath fluid. Data were collected using CytExpert software (Beckman Coulter, USA) and exported to FlowJo version 10.9 software (BD Biosciences, USA) for analysis.

## Microfluidic mother machine device fabrication

Microfluidic mother machine devices were fabricated by pouring a 10:1 (base:curing agent) polydimethylsiloxane (PDMS) mixture (*Łapińska et al., 2023*) into an epoxy mould kindly provided by S. Jun (*Wang et al., 2010*). Each mother machine device contains approximately 6000 bacterial hosting channels with a width, height, and length of 1, 1.5, and 25 µm, respectively. These channels are connected to a main microfluidic chamber with a width and height of 25 and 100 µm, respectively. After degassing the PDMS mixture in a vacuum desiccator (SP Bel-Art, USA) with a two-stage rotary vane pump FRVP series (Fisher Scientific, USA), the PDMS was cured at 70 °C for 2 h and peeled from the epoxy mould to obtain 12 individual chips (*Pagliara et al., 2011*). Fluidic inlet and outlet were achieved using a 0.75 mm biopsy punch (WellTech Labs, Taiwan) at the two ends of the main chamber of the mother machine. The PDMS chip was then washed with ethanol and dried with nitrogen gas, followed by removal of small particles using adhesive tape (3M, USA) along with a rectangular glass coverslip (Fisher Scientific, USA). The microfluidic device was assembled by exposing the PDMS chip and glass coverslip to air plasma treatment at 30 W plasma power for 10 s with the Zepto One plasma cleaner (Diener Electronic, Germany) and then placed in contact to irreversibly bind the PDMS chip and glass coverslip (*Dettmer et al., 2014*). The mother machine was then filled with 5 µL of 50 mg mL$^{-1}$ (752 µM) bovine serum albumin and incubated at 37 °C for 1 h to passivate the charge within the channels after plasma treatment, screening electrostatic interaction between bacterial membranes, PDMS, and glass surfaces (*Cama et al., 2016*).

## Microfluidics-microscopy assay to measure tachyplesin-NBD accumulation and PI staining in individual *E. coli*

A stationary phase *E. coli* culture was prepared as described above. The culture was adjusted to OD$_{600}$ 75 by centrifuging 50 mL of culture in a conical tube at 3220×*g* at room temperature for 5 min in a 5810 R centrifuge (Eppendorf, Germany). The supernatant was removed, and the pellet was resuspended in carbon-free M9 and vortexed. 5 µL of bacterial suspension was then injected into the mother machine device using a pipette and incubated at 37 °C for 30–60 min to allow for filling of ~80% of bacteria hosting channels (*Goode et al., 2021b*). The loaded microfluidic device was then mounted on an Olympus IX73 inverted microscope (Olympus, Japan) connected to a 60x, 1.2 N.A. objective UPLSAPO60XW/1.20 (Olympus, Japan) and a Zyla 4.2 sCMOS camera (Andor, UK) for visual inspection. Each region of interest was adjusted to visualise 23 bacteria hosting channels and a mean of 11 regions of interest were selected for imaging for each experiment via automated stages M-545.USC and P-545.3C7 (Physik Instrumente, Germany) for coarse and fine movements, respectively. Fluorinated ethylene propylene tubing (1/32'×0.0008') was then inserted into both inlet and outlet accesses as previously reported (*Locatelli et al., 2016*). The inlet tubing was connected to a Flow Unit S flow rate sensor (Fluigent, France) and MFCS-4C pressure control system (Fluigent, France) controlled via MAESFLO 3.3.1 software (Fluigent, France) allowing for computerised, accurate regulation of fluid flow into the microfluidic device. A 300 µL h$^{-1}$ flow of carbon-free M9 for 8 min was used to clear the main channel of excess bacteria that had not entered the bacterial hosting channels. Tachyplesin-NBD

(46 µg mL⁻¹ or 18.2 µM) with or without sertraline (30 µg mL⁻¹ or 98 µM) in carbon-free M9 was then flowed through the microfluidic device at 300 µL h⁻¹ for 8 min, before reducing to 100 µL h⁻¹ until 240 min. Carbon-free M9 was then flushed through the device at 300 µL h⁻¹ for 60 min to wash away unaccumulated tachyplesin-NBD. PI staining was performed by flowing PI (1.5 mM) at 300 µL h⁻¹ for 15 min. Brightfield and fluorescence images were captured consecutively every 10 min until the end of the experiment by switching between brightfield and fluorescence mode using LabVIEW (National Instruments, USA). Fluorescence images were captured by illuminating bacteria with a pE-300white broad-spectrum LED (CoolLED, UK) for 0.06 s on the blue excitation band at 20% intensity with a FITC filter for tachyplesin-NBD and green excitation band at 100% intensity with a DAPI/TEXAS filter for PI. The entire experiment was performed at 37 °C in an environmental chamber (Solent Scientific, UK) enclosing the microscope and microfluidics equipment. A step-by-step guide to the process above can be found in *Cama and Pagliara, 2021*.

## Image and data analyses

Images were processed using ImageJ software as previously described (*Smith et al., 2019*). Each individual bacterium was tracked throughout the 240 min tachyplesin-NBD treatment, 60 min carbon-free M9 wash, and 15 min PI staining. To measure tachyplesin-NBD and PI accumulation, a rectangle was drawn around each bacterium in each brightfield image at every time point, obtaining its length, width, and relative position in the microfluidic hosting channel. The same rectangle was then over-laid onto the corresponding fluorescence image to measure the mean fluorescence intensity, which represents the total fluorescence normalised by cell size (i.e. the area covered by each bacterium in the 2D images), thus accounting for variations in tachyplesin-NBD and PI accumulation due to the cell cycle (*Taniguchi et al., 2010*). The same rectangle was then shifted to the nearest adjacent channel that did not contain any bacteria to measure the mean background fluorescence from extra-cellular tachyplesin-NBD in the media. This mean background fluorescence value was subsequently subtracted from the bacterium's mean fluorescence value. To measure tachyplesin-NBD fluorescence profiles, the fluorescence values of a straight horizontal line spanning 20 pixels (left to right) were recorded, then converted and presented in µm. The membrane over cell centre fluorescence ratios were calculated by dividing the fluorescence of the membrane by the fluorescence value of the centre of the cell. The mean membrane fluorescence was obtained using the fluorescence of the seventh and thirteenth pixel, and the cell centre fluorescence using the tenth pixel along the profile measurements. To measure cell sizes of low and high accumulators, the vertical length of the rectangles drawn to measure fluorescence was taken for each cell then converted and presented in µm. All data were then analysed and plotted in Prism version 10.2.0 (GraphPad Software, USA).

Statistical significance tests, including the ordinary one-way ANOVA with Tukey's multiple comparisons tests, paired two-tailed Wilcoxon nonparametric test, unpaired two-tailed nonparametric Mann-Whitney U test, as well as linear and nonlinear (semi-log) regressions were calculated using Prism version 10.2.0 (GraphPad Software, USA).

## Fluorescence-activated cell sorting

Samples for fluorescence-activated cell sorting were generated following the same protocol as flow cytometry accumulation assays. Cells were incubated for 60 min and a wash step carried out before performing a 30x dilution on 500 µL of treated cells for cell sorting. Cell sorting was performed on a BD FACSAria Fusion Flow Cytometer, equipped with a 488 nm (50 mW) laser (BD Biosciences, USA). Drops were generated at 70 psi sheath pressure, 87.0 kHz frequency, and 5.6 amplitude using a 70 µm nozzle. BD FACSFlow (BD Biosciences, USA) was used as sheath fluid and the instrument was set up for sorting with the BD Cytometer Setup & Tracking Beads Kit, and BD FACS Accudrop Beads reagents (BD Biosciences, USA), following the manufacturer's instructions. Cells were sorted based on their fluorescence measured on the 488 nm 530/30 BP channel with photomultiplier tubes (PMT) voltages of FSC: 452, SSC: 302, and 488 nm 530/30 BP: 444, and a threshold value of 200 was applied on the SSC channel to limit background noise. Cell aggregates were excluded by plotting SSC-W against SSC-H. Tachyplesin-NBD-treated cells were sorted into two groups (low and high accumulators) and the untreated control treatment through a control gate. A one-drop sorting envelope with purity rules was used. Approximately 1,000,000 million cells for each group were sorted directly into RNAprotect (Qiagen, Germany) to stabilise RNA in preparation for extraction and into carbon-free

M9 for lipid Folch extractions. An additional combined sample containing a 1:1, *v/v* mixture of sorted low and high accumulators was generated (referred to as 1:1 mix accumulators) to aid and validate downstream transcriptomics analyses. Cells were sorted into carbon-free M9 for cell viability measurements following the same protocol above. Data were collected via BD FACSDiva version 8.0.1 (BD Biosciences, USA) software and analysed using FlowJo version 10.9 software (BD Biosciences, USA). Post-sort analyses revealed high levels of purity and very low occurrences of events measured outside of the sorting gate for sorted cells. Increased events measured in the M9 gate in post-sort samples are due to a reduced concentration of cells in post-sort analyses.

## Cell viability assays

Cell viability was assessed by diluting cells in carbon-free M9, then plating on LB agar plates. Plates were then incubated at 37 °C for 17 h followed by colony-forming unit counts. For flow cytometry accumulation and time-kill assays, sacrificial 30 μL aliquots of cells were obtained from the treatment microcentrifuge tube at their respective time points for dilution and spread plating. For cells sorted via fluorescence-activated cell sorting, low and high tachyplesin-NBD accumulators, and untreated control cells were sorted into carbon-free M9 before dilution and spread plating.

## RNA extraction and sequencing

RNA extractions were performed on cells sorted into RNAprotect (Qiagen, Germany). Enzymatic lysis and proteinase K digestion of bacteria was performed following protocol 4 in the RNAprotect Bacteria Reagent Handbook (Qiagen, Germany). Purification of total RNA from bacterial lysate using the RNeasy Mini Kit (Qiagen, Germany) following protocol 7 in the handbook. Due to the small quantity of initial material from cell sorting, RNeasy MinElute spin columns were used instead of RNeasy Mini spin columns. A further on-column DNase digestion using the RNase-Free DNase set (Qiagen, Germany) was performed following appendix B in the handbook. RNA concentration and quality were assessed using the Agilent High Sensitivity RNA ScreenTape System (Agilent, USA) following the provided protocol. Samples returned a mean RNA concentration of 2.5 ng μL$^{-1}$ and mean RNA integrity number equivalent (RIN$^e$) score of 7.1. rRNA depletion was performed with the Illumina Stranded Total RNA Prep, Ligation with Ribo-Zero Plus kit (Illumina, USA) following the manufacturer's instructions and sequencing carried out on the Illumina NovaSeq 6000 (Illumina, USA). Transcript abundance was quantified using Salmon for each gene in all samples. Differential gene expression was performed with DESeq2 in R software to quantify the log$_2$ fold change in transcript reads (*Love et al., 2014*) for each gene and subsequent principal component analysis using DESeq2 and a built-in R method (prcomp) (*Smith et al., 2018*).

## Cluster and gene ontology analyses

For cluster and gene ontology analysis, differential expression was tested with edgeR (version 3.28.1) (*Robinson et al., 2010*). Predicted log fold changes were positively correlated between replicates; however, a batch effect was detected, and replicate number was retained as a model term. Transcripts with low expression were filtered out of the data via edgeR before fitting the differential expression models. Clustering analysis was performed using the mclust package (version 5.4.7) for R (*Scrucca et al., 2016*). Only transcripts with differential expression FDR <0.05 in at least one cell type were subjected to clustering analysis (*Kraus et al., 2024*). All variance structures were tested for a range of 2–20 clusters, and the minimal, best-fitting model was identified by the Bayes information criterion. Gene ontology enrichment analysis was performed using the clusterProfiler package (version 4.10.0) for R (*Yu et al., 2012*; *Wu et al., 2021*). Enrichment in terms belonging to the 'Biological Process' ontology was calculated for each gene cluster, relative to the set of all genes quantified in the experiment, via a one-sided Fisher exact test (hypergeometric test). p-values were adjusted for false discovery by using the method of Benjamini and Hochberg. Finally, the lists of significantly enriched terms were simplified to remove redundant terms, as assessed via their semantic similarity to other enriched terms, using clusterProfiler's simplify function.

## Transcription factor activity

Relative activities of transcription factors were estimated from differential expression results (generated in edgeR) using VIPER (version 1.2.0) for R (*Alvarez et al., 2016*). Transcription factor regulons

were derived from the full RegulonDB database (version 10.9) (*Santos-Zavaleta et al., 2019*), fixing the likelihood to 1. Only transcripts with differential expression FDR <0.05 in at least one cell type were used to estimate transcription factor activities. VIPER was run with a minimum allowed regulon size of 20 (*minsize = 20*), on the full gene data set (*eset.filter=FALSE*), and was set to normalised enrichment scores (*nes = TRUE*).

## Lipid extraction and lipidomic analysis

Lipid extractions were performed following a modified version of the Folch extraction (*Casula et al., 2023*; *Folch et al., 1951*). Briefly, 100 µL of sample was added to 250 µL of methanol and 125 µL of chloroform in a microcentrifuge tube. Samples were incubated for 60 min and vortexed every 15 min. Then, 380 µL of chloroform and 90 µL of 0.2 M potassium chloride were added to each sample. Cells were centrifuged at 14,000×*g* for 10 min to obtain a lipophilic phase which was transferred to a glass vial and dried under a nitrogen stream.

The lipophilic phase was then resuspended in 20 µL of a methanol:chloroform solution (1:1 *v/v*), then 980 µL of an isopropanol:acetonitrile:water solution (2:1:1, *v/v/v*) was added. Samples were analysed on the Agilent 6560 Q-TOF-MS (Agilent, USA) coupled with the Agilent 1290 Infinity II LC system (Agilent, USA). An aliquot of 0.5 µL for positive ionisation mode and 2 µL for negative ionisation mode from each sample was injected into a Kinetex 5 µm EVO C18 100 A, 150 mm ×2.1 µm column (Phenomenex, USA). The column was maintained at 50 °C at a flow rate of 0.3 mL min⁻¹. For the positive ionisation mode, the mobile phases consisted of (A) acetonitrile:water (2:3 *v/v*) with ammonium formate (10 mM) and (B) acetonitrile:isopropanol (1:9, *v/v*) with ammonium formate (10 mM). For the negative ionisation mode, the mobile phases consisted of (A) acetonitrile:water (2:3 *v/v*) with ammonium acetate (10 mM) and (B) acetonitrile:isopropanol (1:9, *v/v*) with ammonium acetate (10 mM). The chromatographic separation was obtained with the following gradient: 0–1 min 70% B; 1–3.5 min 86% B; 3.5–10 min 86% B; 10.1–17 min 100% B; 17.1–10 min 70% B. The mass spectrometry platform was equipped with an Agilent Jet Stream Technology Ion Source (Agilent, USA), which was operated in both positive and negative ion modes with the following parameters: gas temperature (200 °C); gas flow (nitrogen), 10 L min⁻¹; nebuliser gas (nitrogen), 50 psig; sheath gas temperature (300 °C); sheath gas flow, 12 L min⁻¹; capillary voltage 3500 V for positive, and 3000 V for negative; nozzle voltage 0 V; fragmentor 150 V; skimmer 65 V, octupole RF 7550 V; mass range, 40–1,700 *m/z*; capillary voltage, 3.5 kV; collision energy 20 eV in positive, and 25 eV in negative mode. MassHunter software (Agilent, USA) was used for instrument control.

Low and high tachyplesin-NBD accumulators, and untreated control cells sorted via cell sorting were analysed using quadrupole time-of-flight liquid chromatography/mass spectrometry (Q-TOF-LC/MS). Mass Profiler 10.0 (Agilent, USA) was used to generate a matrix containing lipid features across all samples, then filtered by standard deviation, and normalised by sum calculations. Multivariate statistical analyses were performed on these features using SIMCA software 15.0 (Sartorius, Germany). First, a principal component analysis (PCA) was performed, followed by a partial least square-discriminant analysis (PLS-DA) with its orthogonal extension (OPLS-DA), which was used to visualise differences between low and high tachyplesin-NBD accumulators and untreated control samples.

Tachyplesin-NBD-treated and untreated cells were analysed using ultra-high performance liquid chromatography-quadrupole time-of-flight mass spectrometry (UHPLC-Q-TOF-MS) and detected a mean of 1000 mass spectrometry features in positive and negative ionisation modes. Lipid annotation was performed following COSMOS consortium guidelines (*Salek et al., 2015*) with the CEU Mass Mediator version 3.0 online tool (*Gil-de-la-Fuente et al., 2019*). Lipid Annotator software (Agilent, USA) was then used to identify the lipid classes and their relative abundances (*Figure 3E*). Lipids with variable importance in projection (VIP) scores above 1 were annotated.

## Determination of minimum inhibitory concentration

Exponential phase bacterial cultures were prepared as described above and grown to an OD₆₀₀ of 0.5. 100 µL LB broth was added to all wells of a 96-well microtiter plate (Sarstedt, Germany). Then, 100 µL of a 2048 µg mL⁻¹ (6693.5 µM) sertraline solution was added to the first nine wells of the top row and mixed. Sertraline was then serially diluted to 8 µg mL⁻¹ (26.1 µM) by transferring 100 µL to the next row, mixing thoroughly after each transfer, using a multi-channel pipette. After the final dilution step, 100 µL was discarded to maintain equal volumes across wells. Exponential phase bacterial cultures

were then diluted to yield $10^6$ colony-forming units (CFU) mL$^{-1}$ and added to all wells containing sertraline. Each plate included 9 positive control wells (i.e. bacteria in LB without sertraline), and 15 negative control wells (i.e. LB only, without bacteria or sertraline). Plates were covered with a lid and incubated at 37 °C for 24 h. The MIC of sertraline against *E. coli* BW25113 was determined using the CLARIOstar microplate reader (BMG Labtech, Germany) by measuring OD$_{600}$ after incubation. The MIC was defined as the lowest concentration of sertraline at which no visible growth was observed, as determined by comparison to positive control wells.

## Acknowledgements

The authors thank Lisa Butt and Rob Beardmore, University of Exeter, for providing strains of their Keio collection. This work was supported by the BBSRC through a grant awarded to SP, KTA, and UL (BB/V008021/1), the EPSRC and the MRC through two grants awarded to S.P. (EP/Y023528/1 and MR/Y033892/1). KKL was supported by a Living Systems Institute PhD studentship. SP, KTA, and MATB acknowledge further support from the QUEX Institute. KTA and BMI gratefully acknowledge the financial support of the EPSRC (EP/T017856/1). BEH gratefully acknowledges financial support from the MRC (MR/V009583/1). This project utilised equipment funded by a Wellcome Trust Institutional Strategic Support Fund (WT097835MF), a Wellcome Trust Multi-User Equipment Award (WT101650MA) and a BBSRC LoLa award (BB/K003240/1). The funders had no role in study design, data collection and analysis, decision to publish, or preparation of the manuscript. For the purpose of open access, the author has applied a Creative Commons Attribution (CC BY) licence to any Author Accepted Manuscript version arising from this submission.

## Additional information

### Competing interests

Brandon M Invergo: is affiliated with PIPA LLC. The other authors declare that no competing interests exist.

### Funding

| Funder | Grant reference number | Author |
| --- | --- | --- |
| Biotechnology and Biological Sciences Research Council | BB/V008021/1 | Stefano Pagliara Krasimira Tsaneva-Atanasova Urszula Łapińska |
| Engineering and Physical Sciences Research Council | EP/Y023528/1 | Stefano Pagliara |
| Medical Research Council | MR/Y033892/1 | Stefano Pagliara |
| Engineering and Physical Sciences Research Council | EP/T017856/1 | Krasimira Tsaneva-Atanasova Brandon M Invergo |
| Medical Research Council | MR/V009583/1 | Benjamin E Housden |
| Wellcome Trust | 10.35802/097835 | Aaron R Jeffries |
| Wellcome Trust | 10.35802/101650 | Aaron R Jeffries |
| Biotechnology and Biological Sciences Research Council | BB/K003240/1 | Aaron R Jeffries |
| QUEX Institute | | Stefano Pagliara Krasimira Tsaneva-Atanasova Mark AT Blaskovich |
| Living Systems Institute | PhD studentship | Ka Kiu Lee |

| Funder | Grant reference number | Author |
|--------|------------------------|--------|

The funders had no role in study design, data collection and interpretation, or the decision to submit the work for publication. For the purpose of Open Access, the authors have applied a CC BY public copyright license to any Author Accepted Manuscript version arising from this submission.

## Author contributions

Ka Kiu Lee, Data curation, Investigation, Visualization, Methodology, Writing – original draft, Writing – review and editing; Urszula Łapińska, Data curation, Formal analysis, Supervision, Investigation, Visualization, Methodology, Writing – review and editing; Giulia Tolle, Maureen Micaletto, Bing Zhang, Wanida Phetsang, Anthony D Verderosa, Audrey Farbos, Investigation, Methodology, Writing – review and editing; Brandon M Invergo, Resources, Data curation, Software, Formal analysis, Methodology, Writing – review and editing; Joseph Westley, Data curation, Formal analysis, Methodology, Writing – review and editing; Attila Bebes, Methodology, Writing – review and editing; Raif Yuecel, Resources, Methodology, Writing – review and editing; Paul A O'Neill, Data curation, Software, Formal analysis, Methodology, Writing – review and editing; Aaron R Jeffries, Funding acquisition, Writing – review and editing; Stineke van Houte, Resources, Writing – review and editing; Pierluigi Caboni, Mark AT Blaskovich, Resources, Supervision, Methodology, Writing – review and editing; Benjamin E Housden, Data curation, Supervision, Methodology, Writing – review and editing; Krasimira Tsaneva-Atanasova, Formal analysis, Supervision, Funding acquisition, Writing – review and editing; Stefano Pagliara, Conceptualization, Resources, Data curation, Formal analysis, Supervision, Funding acquisition, Visualization, Writing – original draft, Project administration, Writing – review and editing

## Author ORCIDs

Ka Kiu Lee ⓘ https://orcid.org/0000-0002-2061-3694
Benjamin E Housden ⓘ https://orcid.org/0000-0001-9134-4279
Krasimira Tsaneva-Atanasova ⓘ https://orcid.org/0000-0002-6294-7051
Stefano Pagliara ⓘ https://orcid.org/0000-0001-9796-1956

Reviewer #1 (Public review): https://doi.org/10.7554/eLife.99752.4.sa1
Reviewer #2 (Public review): https://doi.org/10.7554/eLife.99752.4.sa2
Reviewer #3 (Public review): https://doi.org/10.7554/eLife.99752.4.sa3
Author response https://doi.org/10.7554/eLife.99752.4.sa4

# Additional files

## Supplementary files
MDAR checklist

## Data availability
All data generated or analysed during this study are included in the manuscript and supporting files; source data files have been provided for all figures; omics data have been deposited at https://www.ncbi.nlm.nih.gov/bioproject/PRJNA1096674.

The following dataset was generated:

| Author(s) | Year | Dataset title | Dataset URL | Database and Identifier |
|---|---|---|---|---|
| Lee KK, Łapińska U, Tolle M, Micaletto M, Zhang B, Phetsang W, Verderosa AD, Invergo BM, Westley J, Bebes A, Yuecel R, O'Neill PA, Farbos A, Jeffries AR, Houte VS, Caboni P, Blaskovich MAT, Housden BE, Tsaneva-Atanasova K, Pagliara S | 2024 | Heterogeneous efflux pump expression underpins phenotypic resistance to antimicrobial peptides | https://www.ncbi.nlm.nih.gov/bioproject/PRJNA1096674 | NCBI BioProject, PRJNA1096674 |

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

# Appendix 1

**Appendix 1—table 1.** Short list of genes involved in biological processes that are differentially regulated in low compared to high tachyplesin-nitrobenzoxadiazole (NBD) accumulators.

The complete list of genes involved in these processes is reported in *Figure 3—source data 3*.

| Cluster | Regulation in Low Tachyplesin-NBD Accumulators | |
| --- | --- | --- |
| | Up | Down |
| Translation | | |
| 2 | - | *infC* |
| Ribosomal Proteins | | |
| 2 | - | *rplABCDEFKLMNOQRTUWX, rpmACDJ, rpsCDEFHJKMNOS* |
| 3 | - | *rplPSVY, rpmI, rpsABIPQR* |
| RNA Polymerase and Transcription | | |
| 2 | - | *rpoBCS* |
| DNA Replication and Repair | | |
| 2 | - | *gyrAB, hupB, mutY, parC, topA* |
| 3 | - | *mutMT, ruvA* |
| Glucose Metabolism | | |
| 2 | - | *pgm* |
| Persistence Related Genes | | |
| 1 | *spot, dinG* | - |
| 2 | *relA, uvrD, plsB, phoU* | *crp, hupB* |
| 3 | - | *dnaJ, dnaK, lon, soxS* |
| 5 | - | *pspABCDEG* |
| MFS Efflux Pumps | | |
| 1 | *mdfA, mdtG* | - |
| 4 | *emrK, mdtML* | - |
| ABC Transporters | | |
| 1 | *mdlA, sapACDF, yadG* | - |
| 3 | *dppA, oppC, yejA* | *ybhRS, yhhJ* |
| 4 | *ddpABCDF, dppBCDF, hisM, ydcSTUV* | - |
| RND Efflux Pumps | | |
| 1 | *mdtEF* | - |
| 2 | - | *acrB, mdtA* |
| MATE-Type Efflux Pumps | | |
| 2 | - | *mdtK* |
| Outer Membrane Efflux Proteins | | |
| 1 | *tolC* | - |
| Porins | | |
| 3 | - | *ompCF* |

*Appendix 1—table 1 Continued on next page*

*Appendix 1—table 1 Continued*

| Cluster | Regulation in Low Tachyplesin-NBD Accumulators | |
|---|---|---|
| | **Up** | **Down** |
| LPS Biosynthesis, Assembly, Export, and Modification | | |
| 1 | *arnBT, lplT, lptABCD, pgpB, pssA, waaCF, yafK* | - |
| 2 | *gmhA, lpxA, psd* | *clsB, lptEG, eptA* |
| 3 | - | *lpxC, pagP, pgpAC* |
| 4 | *arnF* | - |
| Fatty Acid Biosynthesis | | |
| 1 | - | *fabA* |
| 2 | - | *fabB* |
| Outer Membrane Vesicles | | |
| 1 | *nlpA, waaCF* | - |
| 2 | *lysS* | *pal, tolAB* |
| 3 | - | *degP, mrcB, ompAC* |
| Peptidases | | |
| 1 | *lepB, pepD, yafK* | *prlC* |
| 2 | *pepQ* | *map* |
| 3 | - | *ampH, dacC, lspA, mepM, pepBT* |
| 4 | *ddpX, pbpG* | - |
| 5 | - | - |
| Proteases | | |
| 1 | *degQ, pmbA, ptrB* | - |
| 2 | *ptrA* | *clpA, hflC* |
| 3 | - | *degP, ftsH, loiP, lon, ompT, prc, yccA, ydgD* |
| 5 | - | *htpX* |
| Biofilm and Curli Biosynthesis | | |
| 3 | - | *bssRS, tabA* |
| 4 | *csgCDEFG* | - |

**Appendix 1—table 2.** Lipids and metabolites upregulated in low or high tachyplesin-nitrobenzoxadiazole (NBD) accumulators compared to untreated bacteria.

Lipid or metabolite name, composition, adduct, retention time (RT), theoretical and experimentally measured mass, formula, variable importance in projection (VIP), and concentration in parts per million (Δppm) measured via quadrupole time-of-flight liquid chromatography/mass spectrometry.

| Name | Composition | Adduct | RT | Theoretical mass | Experimental mass | Formula | VIP Score | Δppm |
|---|---|---|---|---|---|---|---|---|
| Lipids and metabolites upregulated in low tachyplesin-NBD accumulators | | | | | | | | |
| LPE 20:4 | - | [M+H] | 1.49 | 502.2928 | 502.2925 | $C_{25} H_{44} NO_7 P$ | 1.46 | −0.59 |
| SM 34:1 | - | [M+H] | 3.40 | 703.5749 | 703.5751 | $C_{39} H_{79} N_2 O_6 P$ | 1.30 | 0.28 |
| PE 32:1 | - | [M+H] | 3.20 | 689.4996 | 689.4997 | $C_{37} H_{72} NO_8 P$ | 1.48 | 0.14 |

*Appendix 1—table 2 Continued on next page*

*Appendix 1—table 2 Continued*

| Name | Composition | Adduct | RT | Theoretical mass | Experimental mass | Formula | VIP Score | Δppm |
|---|---|---|---|---|---|---|---|---|
| PE 36:5 | - | [M+H] | 3.25 | 738.5069 | 738.5067 | $C_{41}H_{72}NO_8P$ | 1.60 | –0.27 |
| PE 38:5 | - | [M+H] | 3.40 | 766.5382 | 766.5380 | $C_{43}H_{76}NO_8P$ | 1.56 | –0.26 |
| SM 34:1 | 18:1; 16:0 | [M-H] | 3.8 | 701.5603 | 701.5603 | $C_{39}H_{79}N_2O_6P$ | 1.97 | 0.28 |
| LPC 18:1 | - | [M-H] | 1.09 | 520.3408 | 520.3406 | $C_{26}H_{52}NO_7P$ | 1.23 | –0.38 |
| **Lipids and metabolites upregulated in high tachyplesin-NBD accumulators** | | | | | | | | |
| Cer 32:33**2:3** | 14:2; 18:1 | [M+H] | 3.32 | 506.4568 | 506.456**45** | $C_{32}H_{59}NO_3$ | 1.37 | –0.78 |
| PC 36:3 | 18:0; 18:3 | [M+H] | 3.74 | 788.5851 | 788.5849 | $C_{44}H_{82}NO_8P$ | 1.42 | –0.25 |
| LPC 22:1 | - | [M+H] | 1.19 | 578.418 | 578.4160 | $C_{30}H_{60}NO_7P$ | 1.46 | –3.45 |
| SM 40:2 | 16:1; 24:1 | [M+H] | 3.48 | 785.6531 | 785.6560 | $C_{45}H_{89}N_2O_6P$ | 1.52 | 3.69 |
| PE 30:0 | 16:0; 14:0 | [M+H] | 3.08 | 664.4912 | 664.4910 | $C_{35}H_{70}NO_8P$ | 1.64 | –0.30 |
| PE 31:0 | 16:0; 15:0 | [M+H] | 3.19 | 678.5069 | 678.5067 | $C_{36}H_{72}NO_8P$ | 1.71 | –0.29 |
| PE 32:2 | 16:1; 16:1 | [M+H] | 3.21 | 688.4912 | 688.4910 | $C_{37}H_{70}NO_8P$ | 1.63 | –0.29 |
| LPC 18:1 | - | [M-H] | 1.09 | 520.3408 | 520.3403 | $C_{26}H_{52}NO_7P$ | 1.91 | –0.96 |
| SM 34:1 | 18:1; 16:0 | [M-H] | 3.8 | 701.5603 | 701.5601 | $C_{39}H_{79}N_2O_6P$ | 1.97 | –0.28 |
| PC 35:0 | - | [M-H] | 2.49 | 774.6018 | 774.6016 | $C_{43}H_{86}NO_8P$ | 1.26 | –0.25 |
| PG 38:7 | - | [M-H] | 1.63 | 791.4868 | 791.4861 | $C_{44}H_{73}O_{10}P$ | 1.63 | –0.88 |
| PS 37:7 | 20:5; 17:2 | [M-H] | 5.69 | 790.4664 | 790.4667 | $C_{43}H_{70}NO_{10}P$ | 1.19 | 0.37 |

