## [Editor Report · eLife Assessment]

This **important** study by Lee et al. explores the heterogeneous response of non-growing bacteria to the antimicrobial peptide (AMP) tachyplesin. The authors identify a subpopulation of cells that evade lethal damage by limiting the intracellular accumulation of a fluorescently labeled tachyplesin analog. The study provides **compelling** evidence that reduced drug accumulation underlies the decreased susceptibility of this subpopulation to the AMP. The molecular basis of this phenotype is well supported by the data.

---

## [Referee Report · Reviewer #1 (Public review)]

Summary:

This work contributes several important and interesting observations regarding the heterotolerance of non-growing *Escherichia coli* and *Pseudomonas aeruginosa* to the antimicrobial peptide tachyplesin. The primary mechanism of action of tachyplesin is thought to be disruption of the bacterial cell envelope, leading to leakage of cellular contents after a threshold level of accumulation. Although the MIC for tachyplesin in exponentially growing *E. coli* is just 1 ug/ml, the authors observe that a substantial fraction of a stationary phase population of bacteria survives much higher concentrations, up to 64 ug/ml. By using a fluorescently labelled analogue of tachyplesin, the authors show that the amount of per-cell intracellular accumulation of tachyplesin displays a bimodal distribution, and that the fraction of "low accumulators" correlates with the fraction of survivors. Using a microfluidic device, they show that low accumulators exclude propidium iodide, suggesting that their cell envelopes remain largely intact, while high accumulators of tachyplesin also stain with propidium iodide. They show that this phenomenon holds for several clinical isolates of *E. coli* with different genetic determinants of antibiotic resistance, and for a strain of Pseudomonas aeruginosa. However, the bimodal distribution does not occur in these organisms for several other antimicrobial peptides, or for tachyplesin in Klebsiella pneumoniae or Staphylococcus aureus, indicating some degree of specificity in the interaction between AMP and bacterial cell envelope. They next explore the dynamics of the fluorescent tachyplesin accumulation and interestingly show that a high degree of accumulation is initially seen in all cells, but that the "low accumulator" subpopulation manages to decrease the amount of intracellular fluorescence over time, while the "high accumulator" subpopulation continues to increase its intracellular fluorescence. Focusing on increased efflux as a hypothesised mechanism for the "low accumulator" phenotype, based on transcriptomic analysis of the two subpopulations, the authors screen putative efflux inhibitors to see if they can block the formation of the low accumulator subpopulation. They find that both the protonophore CCCP and the SSRI sertraline can block the formation of this subpopulation and that a combination of sertraline plus tachyplesin kills a greater fraction of the stationary phase cells than either agent alone, similar to the killing observed when growing cells are treated with tachyplesin.

Strengths:

This study provides new insight into the heterogeneous behaviours of non-growing bacteria when exposed to an antimicrobial peptide, and into the dynamics of their response. The single-cell analysis by FACS and microscopy is compelling. The results provide a much-needed single-cell perspective on the phenomenon of tolerance to AMPs and a good starting point for further exploration.

Weaknesses:

The authors have substantially improved the clarity of the manuscript and have added additional experiments to probe further the location of the AMP relative to low and high accumulators, and the physiological states of these sub-populations. These experiments strengthen the assertion that low accumulators keep the AMP at the cell surface while high accumulators permit intracellular access to the AMP.

The phenomenon of the emergence of low accumulators, which are phenotypically tolerant to the antimicrobial peptide tachyplesin, is interesting and important even if there is still work to be done to understand the mechanism by which it occurs.

---

## [Referee Report · Reviewer #2 (Public review)]

Summary:

This study reports on the existence of subpopulations of isogenic *E. coli* and P. aeruginosa cells that are tolerant to the antimicrobial peptide tachyplesin and are characterized by accumulation of low levels of a fluorescent tachyplesin-NBD conjugate. The authors then set out to address the molecular mechanisms, providing interesting insights even though the mechanism remains incompletely defined: The work demonstrates that increased efflux may cause this phenotype, putatively together with other changes in membrane lipid composition. The authors further demonstrate that pharmacological manipulation can prevent the generation of tolerance. The authors are cautious in their interpretation, and the claims made are largely justified by the data.

Strengths:

Going beyond the commonly used bulk techniques for studying susceptibility to AMPs, Lee et al. used fluorescent antibiotic conjugates in combination with flow cytometry analysis to study variability in drug accumulation at the single cell level. This powerful approach enabled the authors to expose bimodal drug accumulation patterns that were condition-dependent but conserved across a variety of *E. coli* clinical isolates. Using cell sorting in combination with colony-forming unit assays as well as quantitative fluorescence microscopic analysis in a microfluidics setup, the authors compellingly demonstrate that low accumulators (where fluorescence signal is mostly restricted to the membrane) can survive antibiotic treatment, whereas high accumulators (with high intracellular fluorescence) were killed.

The relevance of efflux for the ‘low accumulator’ phenotype and its survival is convincingly demonstrated by the following lines of evidence: (i) A time-course experiment on tachyplesin-NBD pre-loaded cells revealed that all cells initially were high accumulators, before a subpopulation of cells subsequently managed to reduce signal intensity, demonstrating that the ‘low accumulator’ phenotype is an induced response and not a pre-existing property. I(i) Double-mutants deficient in the delta acrA delta tolC double-KO, which showed reduced levels of low accumulators. Interestingly, ‘low accumulator’ populations were nearly abrogated in bacteria deficient in the qse quorum sensing system, suggesting its centrality for the tachyplesin response. Even though this system may control acrA, the strength of the phenotype may suggest that it may control additional as-of-yet unidenitified factors relevant in the response to tachyplesin. I(ii) treatment with efflux pump inhibitor sertraline and verapamil (even though some caution needs to be taken since it is not perfectly selective, see weakness) prevents generation of low accumulators. The observation that sertraline enhances tachyplesin-based killing is an important basis for developing combination therapies.

The study convincingly illustrates how susceptibility to tachyplesin adaptively changes in a heterogeneous way dependent on the growth phases and nutrient availability. This is highly relevant also beyond the presented example of tachyplesin and similar subpopulation-based adaptive changes to the susceptibility towards antimicrobial peptides or other drugs may occur during infections in vivo and they would likely be missed by standardized in vitro susceptibility testing.

Weaknesses:

Some mechanistic questions regarding tachyplesin accumulation and survival remain. One general shortcoming of the setup of the transcriptomics experiment is that the tachyplesin-NBD probe itself has antibiotic efficacy and induces phenotypes (and eventually cell death) in the ‘high accumulator’ cells. As the authors state themselves, this makes it challenging to interpret whether any differences seen between the two groups are causative for the observed accumulation pattern of if they are a consequence of differential accumulation and downstream phenotypic effects.

---

## [Referee Report · Reviewer #3 (Public review)]

Summary:

This important study shows that stationary phase bacteria survive antimicrobial peptide treatment by switching on efflux pumps, generating low accumulating subpopulations that evade killing-a finding with clear implications for the design of peptide based antibiotics and for researchers studying antimicrobial resistance. The evidence is solid and frequently convincing, as diverse single-cell assays, genetics and chemical inhibition coherently link reduced intracellular peptide to survival, even though a few mechanistic details warrant further exploration.

Strengths:

The authors investigate how *Escherichia coli* (and, to a lesser extent, *Pseudomonas aeruginosa*) survive exposure to the antimicrobial peptide (AMP) tachyplesin. Because resistance to AMPs is thought to rely heavily on non genetic adaptations rather than on classical mutation-based mechanisms, the study focuses on phenotypic heterogeneity and seeks to pinpoint the cellular processes that protect a subset of cells. Using fluorescently labelled tachyplesin, single cell imaging, flow cytometry, transcriptomics, targeted genetics, and chemical perturbations, the authors report that stationary phase cultures harbor two phenotypic states: high accumulating cells that die and low accumulating cells that survive. They further propose and show that inducible efflux activity is the primary driver of survival and show that either efflux inhibition (sertraline, verapamil) or nutrient supplementation prevents the emergence of low accumulators and boosts killing.

The experiments unambiguously reveal that the cells respond to stress heterogeneously, with two distinct subpopulations - one with better survival than the other. This primary phenotype is convincingly shown across various *E. coli* strains, including clinical isolates. The authors probed the underlying mechanism from several angles, with important additional experiments in the revised version that strengthens the original conclusions in several ways. Newly added efflux assays with ethidium bromide, together with proteinase treatment experiments and ΔacrAΔtolC and ΔqseB/qseC mutant data, illustrate that the low-accumulating subpopulation can actively export intracellular compounds. The authors took great care to temper their language to acknowledge other potential alternatives that could explain some of the data such as altered influx, vesicle release or proteolysis, metabolic activity of the cells, an indirect effects of sertraline treatment. Additional metabolic dye measurements confirm that low accumulators are less metabolically active, and a new data on nutrient supplementation shows that forcing growth increases peptide uptake and lethality. The authors clarify the crucial point of where antimicrobial peptides actually bind on the cell within the broader survival mechanism and present their conclusions, along with potential caveats, with commendable clarity.

Weaknesses:

Despite these advances, the contribution of efflux may require more direct evidence to further dissect whether efflux is necessary, sufficient, or contributory. The facts that the key low-efflux mutant still retains a small fraction of survivors and that the inhibitors used may cause other physiological changes leading to higher efflux are still unaccounted for. The lipidomic and vesicle findings, while intriguing, remain descriptive, and direct tests of their functional relevance would further solidify the mechanistic models.

Conclusion:

Even with these limitations, the study provides valuable insight into non genetic resistance mechanisms to AMPs and highlights inducible heterogeneity as a critical obstacle to peptide therapeutics. In a much broader context, this study also underscores the importance of efflux physiology even for those antimicrobials that seemingly would not have intracellular targets.

---

## [Author Response]

The following is the authors’ response to the previous reviews.

**Reviewer 1:**

We would like to thank Reviewer 1 for recognising the importance of our findings on the heterogeneity in bacterial responses to tachyplesin.

(1) A double deletion of acrA and tolC (two out of the three components of the major constitutive RND efflux pump) reduces the appearance of the low accumulator phenotype, but interestingly, the single deletions have no effect, and a well-characterised inhibitor of RND efflux pumps also has no effect. The authors identify a two-component system, qseCB, that appears necessary for the appearance of low accumulators, but this system has pleiotropic effects on many cellular systems, with only tenuous connections to efflux. The selected pharmacological agents that could prevent the appearance of low accumulators do not offer clear insight into the mechanism by which low accumulators arise, because they have diverse modes of action.

We have added that “QseBC, was previously inferred to mediate resistance to a tachyplesin analogue by upregulating efflux genes based on transcriptomic analysis and hyper susceptibility of *ΔqseBΔqseC* mutants[113]”. However, we have also acknowledged that “it is conceivable that the deletion of QseBC has pleiotropic effects on other cellular mechanisms involved in tachyplesin accumulation” and that “it is also conceivable that sertraline prevented the formation of the low accumulator phenotype via efflux-independent mechanisms”.

These amendments are reported on lines 525-527, 532-534 and 539-541 of our revised manuscript.

(2) The transcriptomics data collected for low and high accumulator sub-populations are interesting, but in my opinion, the conclusions that can be drawn from these data remain overstated. It is not possible to make any claims about the total amount of "protein synthesis, energy production, and gene expression" on the basis of RNA-Seq data. The reads from each sample are normalised, so there is no information about the total amount of transcript. Many elements of total cellular activity are post-transcriptionally regulated, so it is impossible to assess from transcriptomics alone. Finally, the transcriptomic data are analysed in aggregated clusters of genes that are enriched for biological processes, for example: "Cluster 2 included processes involved in protein synthesis, energy production, and gene expression that were downregulated to a greater extent in low accumulators than high accumulators". However, this obscures the fact that these clusters include genes that are generally inhibitory of the process named, as well as genes that facilitate the process.

We have now acknowledged that “that our data do not take into account post-transcriptional modifications that represent a second control point to survive external stressors”.

These amendments are reported on lines 534-535 of our revised manuscript.

The raw transcript counts can be found in Figure 3 – Source Data, we had added these data in our previous manuscript as requested by this reviewer.

We would also like to clarify that we have analysed our transcriptomic data via both clustering (i.e. Figure 3) and direct comparison of genes of interest (Table S1) and transcription factors (i.e. genes that are generally inhibitory of the process named, as well as genes that facilitate the process, Figure S12).

Finally, we would like to point out that in our revised manuscript (both this and its previous version), we are stating “Cluster 2 included processes involved in protein synthesis, energy production, and gene expression that were downregulated to a greater extent in low accumulators than high accumulators”. We do not think this is an overstatement, and we do not use these data to make conclusions on the total amount of "protein synthesis, energy production, and gene expression".

(3) The authors have added an experiment to attempt to assess overall metabolic activity in the low accumulator and high accumulator populations, which is a welcome addition. They apply the redox dye resazurin and observe lower resorufin (reduced form) fluorescence in the low accumulator population, which they take to indicate a lower respiration rate. This seems possible, however, an important caveat is that they have shown the low accumulator population to retain substantially lower amounts of multiple different fluorescent molecules (tachyplesin-NBD, propidium iodide, ethidium bromide) intracellularly compared to the high accumulator population. It seems possible that the low accumulator population is also capable of removing resazurin or resorufin from the intracellular space, regardless of metabolic rate. Indeed, it has previously been shown that efflux by RND efflux pumps influences resazurin reduction to resorufin in both P. aeruginosa and *E. coli*. By measuring only the retained redox dye using flow cytometry, the results may be confounded by the demonstrated ability of the low accumulator population to remove various fluorescent dyes. More work is needed to strongly support broad conclusions about the physiological states of the low and high accumulator populations. The phenomenon of the emergence of low accumulators, which are phenotypically tolerant to the antimicrobial peptide tachyplesin, is interesting and important even if there is still work to be done to understand the mechanism by which it occurs.

We have now clarified that these assays were performed in the presence of 50 μM CCCP and that “CCCP was included to minimise differences in efflux activity and preserve resorufin retention between low and high accumulators, though some variability in efflux may still persist”. We have now added this information on lines 401-406. This information was only present in the caption of Figure S16 of our previous version of this manuscript.

We agree with the reviewers that more work needs to be done to fully understand this new phenomenon, and we had already acknowledged in our previous version of this manuscript that other mechanisms could play a role in this new phenomenon, see lines 489-517 of the current manuscript.

**Reviewer 2:**

We would like to thank the reviewer for recognising that all their previous comments have now been satisfactorily addressed.

(1) Some mechanistic questions regarding tachyplesin-accumulation and survival remain. One general shortcoming of the setup of the transcriptomics experiment is that the tachyplesin-NBD probe itself has antibiotic efficacy and induces phenotypes (and eventually cell death) in the ‘high accumulator’ cells. As the authors state themselves, this makes it challenging to interpret whether any differences seen between the two groups are causative for the observed accumulation pattern of if they are a consequence of differential accumulation and downstream phenotypic effects.

We agree with the reviewer and we had explicitly acknowledged this possibility on lines 281-285 (of the previous and current version of this manuscript).

(2) The statement ‘Moreover, we found that the fluorescence of low accumulators decreased over time when bacteria were treated with 20 μg mL’ is, in my opinion, not supported by the data shown in Figure S4C. That figure shows that the abundance of ‘low accumulator’ cells decreases over time. Following the rationale that protease K treatment may cleave surface associated/ extracellular tachyplesin-NDB, this should lead to a shift of ‘low accumulator’ population to the left, indicating reduced fluorescence intensity per cell. This is not so case, but the population just disappears. However, after 120 min of treatment more cells appear in the ‘high accumulator’ state. This result is somewhat puzzling.

We agree with the reviewer that our previous discussion of this data could have been misleading. We have now reworded this part of the text as follows: “We found that the fluorescence of high accumulators did not decrease over time when tachyplesin-NBD was removed from the extracellular environment and bacteria were treated with 20 μg mL^-1^ (0.7 μM) proteinase K, a widely-occurring serine protease that can cleave the peptide bonds of AMPs [43–45] (Figure S4B and C). These data suggest that tachyplesin-NBD primarily accumulates intracellularly in high accumulators”.

It is conceivable that extended exposure to proteinase K (i.e. we see a decrease in the abundance of low accumulators after 90 min treatment with proteinase K) increased the permeability to tachyplesin-NBD of low accumulators, allowing tachyplesin-NBD to move from either the extracellular space or the membrane to the cell interior. However, we do not have data to prove this point.

Therefore, we have now removed our claim that the data obtained using proteinase K suggest that tachyplesin-NBD accumulates primarily in the membranes of low accumulators. We believe that our two separate microscopy analyses provide more direct, stronger and less ambiguous evidence that tachyplesin-NBD accumulates primarily in the membranes of low accumulators.

(3) The authors used the metabolic dye resazurin to measure the metabolic activity of low vs. high accumulators. I am not entirely convinced that the lower fluorescence resorufin fluorescence in tachyplesin-NBD accumulators really indicates lower metabolic activity, since a cell's fluorescence levels would also be affected by the cellular uptake and efflux. It appears plausible that the lower resorufin-fluorescence may result from reduced accumulation/increased efflux in the ‘low-tachyplesin NBD´ population.

We have now clarified that these assays were performed in the presence of 50 μM CCCP and that “CCCP was included to minimise differences in efflux activity and preserve resorufin retention between low and high accumulators, though some variability in efflux may still persist.” We have now added this information on lines 401-406. This information was only present in the caption of Figure S16 of our previous version of this manuscript.

(4) P8 line 343. The text should refer to Figure. 13B, instead of 14B

We have now changed the text accordingly on line 337.

**Reviewer 3:**

We would like to thank the reviewer for recognising that we have done a very impressive job in taking care of their comments.

(1) Despite these advances, the contribution of efflux may require more direct evidence to further dissect whether efflux is necessary, sufficient, or contributory. The facts that the key low efflux mutant still retains a small fraction of survivors and that the inhibitors used may cause other physiological changes leading to higher efflux are still unaccounted for. The lipidomic and vesicle findings, while intriguing, remain descriptive, and direct tests of their functional relevance would further solidify the mechanistic models.

We agree with the reviewers that more work needs to be done to fully understand this new phenomenon, and we had already acknowledged in our previous version of this manuscript that other mechanisms could play a role in this new phenomenon, see lines 489-517 of the current manuscript.